# ENHANCED VISUAL INSTRUCTION TUNING FOR TEXT-RICH IMAGE UNDERSTANDING

## ABSTRACT

Instruction tuning enhances the capability of Large Language Models (LLMs) to interact with humans. Furthermore, recent instruction-following datasets include images as visual input, collecting responses for image-based instructions. However, current visual instruction-tuned models cannot comprehend textual details within images well. This work enhances the current visual instruction tuning pipeline with text-rich images (*e.g.*, movie posters, book covers, etc.). Specifically, we first used publicly available OCR tools to collect results on 422K text-rich images from the LAION dataset. Furthermore, we prompt text-only GPT-4 with recognized text and image captions to generate 16K conversations, each containing question-answer pairs for text-rich images. By combining our collected data with previous multimodal instruction-following data, our model, **LLaVAR**, substantially improves the capability of the LLaVA model on text-based VQA datasets (up to 20% accuracy improvement). The GPT-4-based instruction-following evaluation also demonstrates the improvement of our model on both natural images and text-rich images. Through qualitative analysis, LLaVAR shows promising interaction skills ( *e.g.*, reasoning, writing, and elaboration) with humans based on the latest real-world online content that combines text and images. We make our code/data/models publicly available.

## 1 INTRODUCTION

Instruction tuning (Ouyang et al., 2022; Chung et al., 2022) improves generalization to unseen tasks by formulating various tasks into instructions. Such open-ended question-answering capability fosters the recent chatbot boom since ChatGPT. Recently, visual instruction-tuned models (Liu et al., 2023a; Li et al., 2023a; Li, 2023) further augment conversation agents with visual encoders such as CLIP-ViT (Dosovitskiy et al., 2020; Radford et al., 2021), enabling human-agent interaction based on images. However, possibly due to the dominance of natural images in training data (e.g., Conceptual Captions (Changpinyo et al., 2021) and COCO (Lin et al., 2015)), they struggle with understanding texts within images (Liu et al., 2023c). However, textual understanding is integral to visual perception in everyday life.

Fortunately, tools such as Optical Character Recognition (OCR, Mori et al., 1992) allow us to recognize text in images. One naive way to utilize this is to add recognized texts to the input of visual instruction-tuned models (Gao et al., 2023). However, such approach significantly increases the computation (longer context lengths), and might not fully leverage the encoding capability of visual encoders. To this end, we propose to enhance the end-to-end visual instruction-tuned model by collecting instruction-following data that require understanding texts within images.

Specifically, we first collect 422K noisy instruction-following data using text-rich[1] images by combining manually written instructions (e.g., "Identify any text visible in the provided image.") and the OCR results. Such large-scale noisy-aligned data effectively improve feature alignment between visual features and the language decoder. Furthermore, we prompt text-only GPT-4 (OpenAI, 2023) with OCR results and image captions to generate 16K conversations, where each conversation can be multiple turns of question & answer pairs, as high-quality instruction-following examples. This

---

[1]In this work, we use the phrase "text-rich images" to describe images with text in them, such as posters and book covers. In contrast, we refer to images without text as "natural images".

***OCR1:*** Peep- Through Book Lets Go Under the Seal Petr Horacek
***OCR2:*** A Peep-Through Book Let's Go Under the Sea! Petr Horacek
***Image Captioning:*** a picture of a yellow submarine with a boy in it

**Text-Only GPT-4**

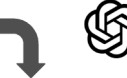

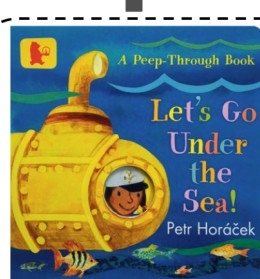

Question: What is the name of the book mentioned in the image?
Answer: The name of the book mentioned in the image is "Let's Go Under the Sea" by Petr Horacek.
Question: What type of book is it?
Answer: It is a "Peep Through Book," which usually features die-cut holes to encourage interaction and exploration, making the reading experience more engaging for children.
Question: Can you describe the scene shown in the image?
Answer: The scene shown in the image features a yellow submarine with a boy in it, which correlates with the theme of the book, "Let's Go Under the Sea."

Figure 1: The process of collecting high-quality instruction-following data.

process requires GPT-4 to de-noise the OCR results and develop specific questions to create complex instructions based on the input (Figure 1).

To evaluate the effectiveness of the collected data, we use noisy and high-quality examples to augment the pretraining and fine-tuning stages of LLaVA (Liu et al., 2023a) accordingly. We name our model **LLaVAR**, signifying the LLaVA (Large Language and Vision Assistant) that can **R**ead. Compared to the original LLaVA, we also conducted experiments scaling the input resolution from $224^2$ to $336^2$ to better encode small textual details. Empirically, we report the results on four text-based VQA datasets following the evaluation protocol from Liu et al. (2023c). Moreover, we apply GPT-4-based instruction-following evaluation to 30 natural images from COCO (Lin et al., 2015; Liu et al., 2023a) and 50 text-rich images from LAION (Schuhmann et al., 2022). We also provide qualitative analysis (e.g., on posters, website screenshots, and tweets) to test more complex instruction-following skills.

To sum up, our contributions are as follows:

- We collect 422K noisy instruction-following data and 16K high-quality instruction-following data. Both are shown to be effective in augmenting visual instruction tuning.
- Our model, LLaVAR, significantly enhances text understanding within images while slightly improving the model's performance on natural images.
- The enhanced capability enables our model to provide end-to-end interactions based on various forms of online content that combine text and images.
- We open source the training and evaluation data together with the model checkpoints.

## 2 RELATED WORK

**Instruction Tuning**  Following natural language instructions is the key capability for an agent to interact with real-world users. Instruction tuning starts from collecting human-preferred feedback for human written instructions (Ouyang et al., 2022) or formulating multi-task training in a multi-task instruction-following manner (Chung et al., 2022; Wang et al., 2022b). However, large, capable instruction-tuned models are usually closed-sourced and serve as commercial APIs only. Recently, Alpaca (Wang et al., 2022a; Taori et al., 2023), Vicuna (Chiang et al., 2023), and Baize (Xu et al., 2023) start the trend of generating high-quality instruction-following data based on LLMs such as GPT-3.5 / ChatGPT / GPT-4 and finetuning the open source LLaMA model (Touvron et al., 2023). However, evaluating the ability to follow instructions remains a challenge. While GPT-4 has demonstrated superior evaluation capabilities (Liu et al., 2023b), there are still a number of concerns, such as bias toward response length (Xu et al., 2023) and lack of robustness to the order of examples (Wang et al., 2023). Following Chiang et al. (2023); Liu et al. (2023a); Dubois et al. (2023), we use GPT-4-based instruction-following evaluation in this work.

**Multimodal Instruction Tuning**   Recently, instruction tuning has been expanded to the multi-modal setting, including image, video (Zhang et al., 2023b; Maaz et al., 2023), and audio (Huang et al., 2023; Zhang et al., 2023a). For image-based instruction tuning, MiniGPT-4 (Zhu et al., 2023) employs ChatGPT to curate and improve detailed captions for high-quality instruction-following data. LLaVA (Liu et al., 2023a) generates multimodal instruction-following data by prompting text-only GPT-4 with captions and object's bounding boxes. LLaMA-Adapter (Zhang et al., 2023c; Gao et al., 2023) uses COCO data for text-image feature alignment and utilizes textual data only for instruction tuning. mPLUG-owl (Ye et al., 2023) combines more than 1000M image-text pairs for pretraining and a 400K mixture of text-only/multimodal instruction-following data for finetuning. However, according to Liu et al. (2023c), most of these models struggle to accomplish tasks requiring OCR capability. InstructBLIP (Dai et al., 2023) transforms 13 vision-language tasks (including OCR-VQA (Mishra et al., 2019)) into the instruction-following format for instruction tuning. Cream (Kim et al., 2023) applies multi-task learning that includes predicting masked texts in images. A more comprehensive survey can be found in Li et al. (2023b). In this work, we select LLaVA as our baseline, which is the most data-efficient and powerful model, and demonstrate the effectiveness of our proposed pipeline.

# 3   DATA COLLECTION

Starting from the LAION-5B (Schuhmann et al., 2022) dataset [2], our goal is only to keep images that are text-rich. Considering that documents usually contain plenty of text, we first obtained a binary classification dataset by combining natural images and document data. Subsequently, we trained an image classifier using a DiT (Li et al., 2022)-base backbone, which was fine-tuned on the RVL-CDIP dataset (Harley et al., 2015). Hopefully, such a classifier can predict whether an image contains text or not. We first build a subset by selecting images with a predicted probability greater than 0.8 while also satisfying $p(\text{watermark}) < 0.8$ and $p(\text{unsafe}) < 0.5$ [3]. The derived subset is noisy due to the limitation of the classifier. To further clean up the data and incorporate human judgment,

we randomly sampled 50K images and clustered them into 100 clusters based on `CLIP-ViT-B/32` visual features. After inspecting the clustering results, we carefully select 14 clusters (see Figure 10 in the Appendix for examples) containing diverse text-rich images ranging from posters, covers, advertisements, infographics, educational materials, and logos. The cluster model is then used as the filter to collect images for constructing our instruction-following examples. As a reference, we provide a CLIP (Radford et al., 2021)-based categorization (see Appendix A for details.) to illustrate the distribution of images for both two types of data we collected in Figure 2. We summarize our collected data and LLaVA's data in Table 1.

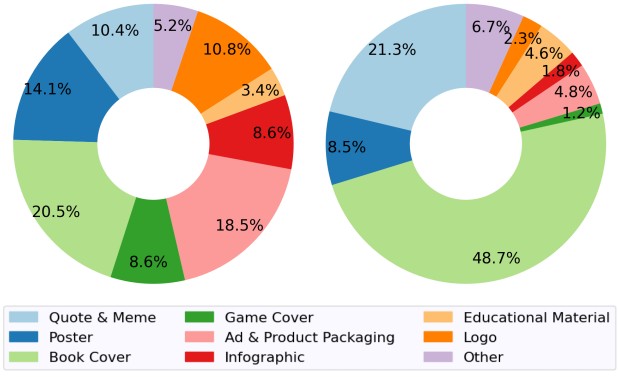

Figure 2: CLIP-based categorization of our collected images. The left refers to images used to collect noisy data, and the right refers to images used in the GPT-4 prompting. Both pie charts are based on 10K sampled images from the corresponding datasets.

**Noisy Instruction-following Data**   Using the clustering model as a filter, we collect 422K deduplicated images that belong to the 14 preferred clusters. To balance the examples from different categories, we keep at most 52K examples for one cluster. We run all images through PaddleOCR [4]. Note that running OCR at the original resolution (e.g.,$1024^2$) might recognize small fonts that are not visible by visual encoders like CLIP ViT (Dosovitskiy et al., 2020; Radford et al., 2021, resolution up to $336^2$). To ensure the recognition of visible fonts while maintaining OCR accuracy, we

---

[2] https://huggingface.co/datasets/laion/laion-high-resolution

[3] Both probabilities are from the LAION dataset's metadata.

[4] https://github.com/PaddlePaddle/PaddleOCR

| Data | Image | Instruction | # Conv | Avg Ins Len | Avg Res Len |
|---|---|---|---|---|---|
| LLaVA pretraining | CC3M | CC3M | 595K | 15.9 | 15.4 |
| $R_{pretraining}$ (Ours) | LAION | PaddleOCR | 422K | 17.2 | 48.8 |
| LLaVA finetuning | COCO | GPT-4 | 158K | 15.9 | 93.1 |
| $R_{finetuning}$ (Ours) | LAION | GPT-4 | 16K | 15.1 | 40.5 |

Table 1: Summary of data statistics. $R_{pretraining}$ and $R_{finetuning}$ denote the additional pre-training / finetuning data we collected. The average instruction and response length are calculated after LLaMA tokenization.

perform OCR on the image after downsampling (the short edge is resized to 384 pixels if longer than that.) to extract the text. Then, based on the geometric relationships between the recognized words, we merge them into paragraphs before concatenating them. As a robust instruction-following model should react similarly to instructions with similar meanings, we reword "Identify any text visible in the provided image." into ten distinct instructions (Table 6 in Appendix). We then create a single-turn conversation for a given image by **(i)** randomly sampling an ***input instruction*** and **(ii)** using recognized texts as the desired ***output response***. Such instruction-following data is noisy because of the relatively limited performance of OCR tools on diverse fonts and colorful backgrounds.

**GPT-4-based Instruction-following Data**  Compared to high-quality instruction-following data, there are mainly two issues for the noisy data collected above. **(i)** Responses should contain organized sentences instead of raw OCR results with missing words and grammar errors. **(ii)** Instructions should be diverse, suitable and specific to the given image instead of monotonously asking for all visible texts. To address these issues, we follow Liu et al. (2023a) to generate instruction-following data by prompting text-only GPT-4 (OpenAI, 2023) with OCR results and captions.

It is challenging to prompt GPT-4 with fragmented OCR results in a few words to generate non-trivial instructions. To this end, we carefully select 4 of the 14 previously mentioned clusters (the 3rd, 4th, 6th and 9th clusters in Figure 10) to collect images with enough visible and coherent sentences. As shown in Figure 2, such filtering dramatically increases the percentage of book covers and quote images. We randomly selected 4K examples from each cluster (no overlap with images used for noisy instruction-following data), yielding a total of 16K images. Following prior work (Wang et al., 2022a; Taori et al., 2023; Liu et al., 2023a), we provide the visualization of verb-noun pairs for instructions generated by GPT-4 in Appendix Figure 11. For those instructions without a verb-noun pair, we demonstrate the frequency of objects being asked in Appendix Figure 12.

Furthermore, based on the system message and two in-context few-shot examples (shown in Appendix B), we ask GPT-4 to generate conversational data based on OCR results and image captions (Figure 1). The generated questions are used as ***input instructions***, and answers are used as ***output responses***. Concretely, for a given image, we first provide two OCR results from EasyOCR and PaddleOCR, which can complement each other. To illustrate visual elements other than texts within the image, we also provide the result of BLIP-2 image captioning (Li et al., 2023c). To prevent the caption from focusing on the text, we use OCR bounding boxes to mask the text and then use the inpainting (Telea, 2004) to refill the mask before using generation captions. Note that captioning models might suffer from hallucinations (Rohrbach et al., 2018). We mention this unreliability in our system message and ask GPT-4 only to generate questions with sure answers. We leave the generation of more detailed captions (Rotstein et al., 2023; Hu et al., 2022) for future work.

## 4 MODEL ARCHITECTURE AND TRAINING

**Architecture**  In most of our study, we use the same model architecture as LLaVA. For the visual encoder $V$, we use `CLIP-ViT-L/14` for $224^2$ resolution and `CLIP-ViT-L/14-336` for $336^2$ resolution. The grid features before the last transformer layer are then transformed into the word embedding space of the language decoder through a trainable projection matrix $W$. We use Vicuna-13B (Chiang et al., 2023), a LLaMA-based (Touvron et al., 2023) instruction-tuned language model, as the language decoder $D$ except the ablation study in Table 4.

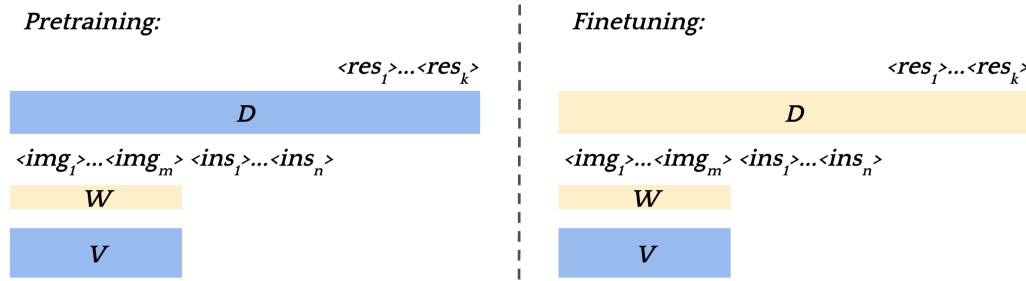

Figure 3: The model training process for the visual encoder $V$, projection matrix $W$, and language decoder $D$. **Blue blocks** denote frozen modules and *yellow blocks* denote trainable modules. The training input is image tokens ($$) and instruction tokens ($$), while the target is response tokens ($<res>$).

In Section 5.1 and Appendix H, we extend the current architecture by adding an extra high-resolution (high-res) visual encoder. Such a high-res encoder outputs thousands of patch features, which means that the transformed features and instruction tokens cannot fit in the context length of the language decoder. To this end, we propose to add cross-attention modules to the decoder, which attend to key-value pairs transformed from the high-res patch features.

**Training**  We follow the two-stage training design of LLaVA (Figure 3). The training objectives of both stages are the same: generate ***output responses*** ($<res>$) for the ***input instructions*** ($$). The transformed image tokens ($$) are added before or after the first input instruction. **(i)** During the first pre-training stage, only the projection matrix $W$ is trained for feature alignment. Since the decoder $D$ is frozen, training tolerates noisy data. In the pre-training stage, we combine the 595K pre-training data from LLaVA with our 422K noisy instruction-following data. **(ii)** Both the projection matrix $W$ and the language decoder $D$ are trained during the finetuning stage, where we merge our 16K instruction-following data into the 158K instruction-following data from LLaVA as the training set. Note that the visual encoder is frozen throughout the training period, which might restrict text recognition performance, as CLIP is trained for general-purpose text-image alignment. Better choices of the visual encoder (Tschannen et al., 2022) or CLIP-ViT finetuning (Ye et al., 2023) may further benefit the visual understanding capability, which we leave for future work.

## 5  EXPERIMENTS

We use the same training hyperparameters as LLaVA[5], except that **(i)** We set the maximum sequence length to 1024 during pre-training. **(ii)** We first pad any given image to a square shape before resizing it to the desired input size, preventing some image content from cropping during preprocessing. For both resolutions ($224^2$, $336^2$), we reproduce the original LLaVA for a fair comparison. The GPT-4 model used in this work refers to the `gpt-4-0314` version, while the cost to collect finetuning data is around $300. The temperature used to sample GPT-4 is set to 1.0 for the generation of training data, 0.7 for the generation of evaluation data, and 0.2 for the evaluation based on GPT-4. All experiments are run on NVIDIA A100 80GB GPUs. During the evaluation, the temperature used to sample from our model is set at 0.9 for text-based VQA, 0.7 for GPT-4-based instruction-following evaluation, and 0.2 for other qualitative demonstrations.

### 5.1  QUANTITATIVE ANALYSIS

**Text-based VQA**  Following the evaluation protocol in Liu et al. (2023c), we evaluate the performance of LLaVAR on four text-based VQA datasets: ST-VQA (Furkan Biten et al., 2019), OCR-VQA (Mishra et al., 2019), TextVQA (Singh et al., 2019), and DocVQA (Mathew et al., 2020), representing various domains (see Appendix C for more details and Appendix E for more datasets). We present the results of the baseline models and our models in Table 2. Note that InstructBLIP

---
[5]https://github.com/haotian-liu/LLaVA

| | Res. | ST-VQA | OCR-VQA | TextVQA | DocVQA |
|---|---|---|---|---|---|
| BLIP-2 (2023c) † | | 21.7 | 30.7 | 32.2 | 4.9 |
| OpenFlamingo (2023) † | | 19.3 | 27.8 | 29.1 | 5.1 |
| MiniGPT4 (2023) † | $224^2$ | 14.0 | 11.5 | 18.7 | 3.0 |
| LLaVA (2023a) † | | 22.1 | 11.4 | 28.9 | 4.5 |
| mPLUG-Owl (2023) † | | 29.3 | 28.6 | 40.3 | 6.9 |
| LLaVA ‡ | $224^2$ | 24.3 | 10.8 | 31.0 | 5.2 |
| LLaVAR | | 30.2  (+5.9) | 23.4  (+12.6) | 39.5  (+8.5) | 6.2  (+1.0) |
| LLaVA ‡ | $336^2$ | 28.9 | 11.0 | 36.7 | 6.9 |
| LLaVAR | | 39.2  (+10.3) | 23.8  (+12.8) | 48.5  (+11.8) | 11.6  (+4.7) |

Table 2: Results (accuracy %) on text-based VQA. We use † to refer to the results obtained from Liu et al. (2023c) and ‡ to refer to our reproduced results. The accuracy metric used by Liu et al. (2023c) only counts for whether the ground truth appears in the response. For more metrics, please refer to Appendix D.

| | ST-VQA | OCR-VQA | TextVQA | DocVQA |
|---|---|---|---|---|
| (1) LLaVA | 28.9 | 11.0 | 36.7 | 6.9 |
| (2) LLaVA + $R_{pretraining}$ | 36.7 | 26.1 | 46.5 | 9.6 |
| (3) LLaVA + $R_{finetuning}$ | 34.1 | 21.6 | 43.6 | 9.5 |
| (4) LLaVA + $C_{pretraining}$ | 35.4 | 27.0 | 45.6 | 9.2 |
| (5) LLaVA + $N_{finetuning}$ | 34.1 | 25.9 | 43.3 | 10.2 |
| (6) LLaVAR | 39.2 | 23.8 | 48.5 | 11.6 |

Table 3: Ablation Study on pretraining/finetuning data. All results are from $336^2$-based models. $R_{pretraining}$ and $R_{finetuning}$ denote the extra pretraining/finetuning data we collected. $C_{pretraining}$ refers to using captions instead of OCR results as responses during pretraining. $N_{finetuning}$ refers to using written questions + raw OCR results instead of GPT-generated QA for finetuning.

includes OCR-VQA in its training sets, making it incomparable with our settings. In both resolution settings and all four datasets, LLaVAR substantially improves the LLaVA baseline, demonstrating that our collected data can bring about a robust improvement. Furthermore, the improvement is more significant in $336^2$ resolution compared to $224^2$, indicating that the collected data might bring a greater improvement at even higher resolutions. Our best model, $336^2$-based LLaVAR, performs best in 3 out of 4 evaluated datasets. Note that this is not a fair comparison. Some key factors include different language decoders, resolutions, and magnitudes of text-image training data. We provide more discussions on the comparison with mPLUG-Owl and the result of finetuning mPLUG-Owl using our data in Appendix F.

**Ablation Study on pretraining/finetuning data**  We report the result in Table 3 and Figure 4. **(i)** Based on variants (2) and (3), we find that the collected data can benefit the pretraining stage ($R_{pretraining}$) and finetuning stage ($R_{finetuning}$) separately while being complementary to each other in most cases [6]. More importantly, enhancing the pretraining stage alone achieves the second-best overall performance, indicating the potential to boost textual detail understanding without dependence on GPT-4-generated high-quality data. **(ii)** Using pretraining images, we obtain $C_{pretraining}$ by replacing the pretraining instructions with questions & captions, the same pattern as LLaVA. As variant (4) is not as good as (2), we can conclude that OCR is more advantageous than captions. **(iii)** We further validate the value of GPT-4 generated data by generating noisy finetuning data ($N_{finetuning}$), similar to pretraining data. Variant (5) achieves comparable accuracy as variant (3). However, as shown in Figure 4, such noisy finetuning data hurts the instruction-following capability: (5) responds with all recognized texts while ignoring the questions.

---

[6] Since the metric only consider the recall, it might favor variant (2)(4)(5) due to their longer outputs.

| | CLIP Res. | Extra Enc. | $R_{\text{pretraining}}$ | ST-VQA | OCR-VQA | TextVQA | DocVQA |
|---|---|---|---|---|---|---|---|
| (a) | $224^2$ | ✗ | Low | 28.9 | 25.6 | 37.8 | 6.2 |
| (b) | $336^2$ | ✗ | Low | 37.4 | 31.0 | 45.7 | 11.4 |
| (c) | $224^2$ | ✗ | High | 28.9 | 24.9 | 35.8 | 6.2 |
| (d) | $336^2$ | ✗ | High | 36.9 | 30.6 | 46.5 | 11.7 |
| (e) | $224^2$ | ✓ | Low | 31.8 | 30.9 | 41.2 | 8.1 |
| (f) | $224^2$ | ✓ | High | 33.5 | 29.8 | 42.2 | 10.2 |
| (g) | $224^2$ | ✓ | High (2×) | 35.8 | 30.7 | 45.6 | 15.3 |

Table 4: Ablation Study on encoders and image resolution. "Extra Enc." refers to whether we use `Pix2Struct-base` (Lee et al., 2022) as an extra high-res visual encoder. "$R_{\text{pretraining}}$" refers to images used for pretraining (before preprocessing) and the collection of OCR results, where "Low" refers to the resized images (the same as the rest of this paper), "High" refers to the original high-res images, and "2×" means doubling the number of examples using the same criteria. All results from this table use Vicuna-7B, as adding the extra encoder requires more computation and memory.

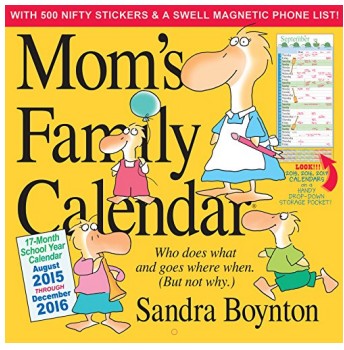

| **Question**: Who wrote this book? |
|---|
| GT     Sandra Boynton |
| (1)    The book was written by Sandra Boynton. |
| (2)    The book was written by Sandra Boyton. It is a children's book that features a family calendar with an accompanying activity guide. The book provides information (...) The cover of the book displays a picture of the family from the calendar, which helps to visually represent the topic of the book. |
| (3)    The book is written by Sandra Byington. |
| (4)    The book was written by Sandra Boynton. |
| (5)    With 500+ Nity StickerS &aSwEllMagnetic Phone List! Moms 291573016171800 FAMII (Pocket) Calendar (...) 2016 SANDRA BOYNTON |
| (6)    The book was written by Sandra Boynton. |

Figure 4: Ablation study based an example from OCR-VQA. GT refers to ground truth, and (1) - (6) are different model variants from Table 3. We replace the excessively long response with (...).

**Ablation Study on encoders/image resolution**    While keeping finetuning data the same, we report the quantitative results of adding an extra visual encoder and varying the pretraining data in Table 4. **(i)** Take `Pix2Struct-base` as an example, we find that adding an extra high-res visual encoder with cross-attention indeed improves the performance ((g) vs. (a)), especially achieving the best zero-shot performance on DocVQA (15.3% accuracy). The performance gain on other datasets is relatively limited, probably due to the extra encoder we use being pretrained on web pages instead of natural images. On the other hand, the performance of (e) and (f) remains poor, without doubling the number of high-res examples in $R_{\text{pretraining}}$. Given the larger number of parameters initialized in the cross-attention module, they may be underfitting when trained on the same data as the projection matrix $W$ (e.g., (e) vs. (b)), similar to the finding in Zeng et al. (2023). **(ii)** Considering (c) vs. (a) and (d) vs. (b), while the images are resized to the same size after preprocessing, high-res OCR results turn out to be not necessarily better than the low-resolution version, suggesting the capability of the visual encoder is almost saturated in (a) and (b). For more details and results on the extra high-res encoder, please refer to Appendix H.

**GPT-4-based instruction-following evaluation**    We also report the GPT-4 evaluation results on instruction-following questions in Table 5. **(i) Natural Images**: 90 questions based on 30 COCO validation images from Liu et al. (2023a), including three aspects: conversation, detail description, and complex reasoning. This aims to test whether our collected data will hurt, maintain, or improve the model's performance on natural images. First of all, using a higher resolution brings improvement (+2.9) in the performance of detail description, which is intuitive. Furthermore, LLaVAR achieves a better trade-off and increases the performance of all three aspects (+1.6 on average).

|                  | Res      | Conversation | Detail | Complex | Read |
|------------------|----------|--------------|--------|---------|------|
| LLaVA (Original) | $224^2$  | 83.1         | 75.3   | 96.5    | -    |
| LLaVA            | $336^2$  | 83.9         | 78.2   | 95.3    | 87.9 |
| LLaVAR           | $336^2$  | 84.5         | 78.9   | 96.5    | 91.7 |

Table 5: Relative scores (w.r.t. text-only GPT-4) for instruction-following questions, where the first three dimensions are based on natural images, the last dimension ("Read") is based on text-rich images. In the first row, we show the original results ($224^2$-based) fetched from Liu et al. (2023a). We report our reproduced LLaVA on $336^2$ resolution for a fair comparison.

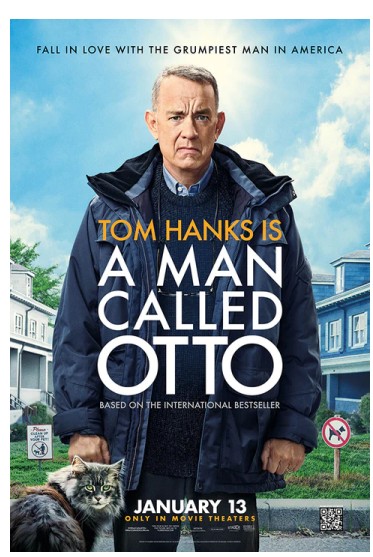

*H:* When will this movie be released?
*L:* The movie "The Call of the Ottoman" starring Daniel Craig is expected to be released on February 14th, 2023.
*R:* The movie will be released on January 13.
*H:* Who starred in the movie?
*L:* The movie stars Tom Cruise and Cobie Smulders.
*R:* Tom Hanks starred in the movie.
*H:* What is this movie about?
*L:* The movie "A Man Called Ove" is a comedy-drama film directed by Hannes Holm. (...)
*R:* The movie is about Tom Hanks, who is called Ottol, and it is an international bestseller based on a true story. (...)
*H:* Can you describe the given image?
*L:* The image is a movie poster featuring a man and a cat. The man, who appears to be a character from the movie "The Ugly Truth," (...)
*R:* The image features a man standing in front of a house with a cat beside him.
*H:* Who directed the film?
*L:* The film was directed by Florian Henckel von Donnersmarck.
*R:* The film was directed by Tom Hanks, who made his directorial debut with this movie.

Figure 5: Qualitative Comparison. We color the human inputs (*H*) in blue, LLaVA's responses (*L*) in pink, and LLaVAR's responses (*R*) in green. Both are $336^2$-based models. We replace the excessively long response with (...).

More details are in Appendix J. **(ii) Text-Rich Images**: Similar to collecting the finetuning data, we leverage 50 text-rich images from LAION to collect instruction-following questions based on OCR results and human-annotated captions. We then collect responses from our trained model and use GPT-4 to calculate the relative score w.r.t GPT-4 responses. We add this as an extra dimension "**Read**" to Table 5, where our model demonstrates a significant (+3.8) improvement. The Appendix provides an example in Table 17.

## 5.2 QUALITATIVE ANALYSIS

We use a recent movie poster [7] to demonstrate the difference between LLaVA and LLaVAR when interacting with humans based on text-rich images. LLaVA, without augmenting textual understanding within images, suffers from hallucination when answering these questions. Some mentioned movies, like "A Man Called Ove" and "The Ugly Truth", are real movies, suggesting that the language decoder is hallucinating its internal knowledge, while the visual encoder cannot encode helpful information. Alternatively, LLaVAR can correctly answer many of the provided questions with **faithful** information, which is clearly grounded in the image. However, some limitations remain, such as the spelling error "ottol" (We provide more statistics related to such spelling errors in Appendix I). Also, the final question asks for information that is not observable from the given poster, where an expected response should express such uncertainty instead of giving concrete answers. Nevertheless, neither model correctly answers the question.

---

[7] https://www.imdb.com/title/tt7405458/

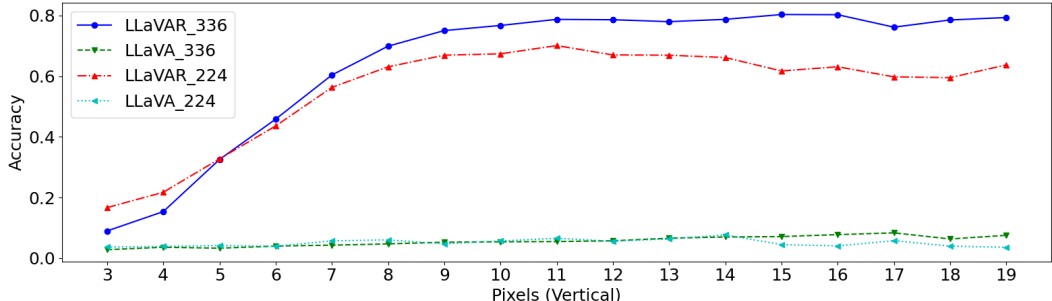

Figure 6: Case study of the recognizable font size, in which the x-axis refers to the height of ground truth answers in the image and the y-axis stands for the answer accuracy of models. We plot the results for both $224^2$-based models and $336^2$-based models.

## 5.3 CASE STUDY: RECOGNIZABLE FONT SIZE

We first collect 825 examples from OCR-VQA, which have answers directly presented in the image and are detectable by the OCR tool. By rescaling the images, we test the model's performance in answering these questions while the vertical heights of answers range from 3 pixels to 19 pixels. We report the result in Fig 6. **(i)** For the baseline model LLaVA, it struggles to provide correct answers in all scenarios, for both $224^2$-based and $336^2$-based versions. **(ii)** Our model LLaVAR achieves significantly better results in all scales. We observe a threshold for recognizable texts for both $224^2$-based and $336^2$-based versions as the accuracy sharply decreases when the height is smaller than 7 pixels. More interestingly, the $224^2$-based version achieves better performance on small texts with 3 pixels height while the $336^2$-based achieves better performance on large texts with more than 7 pixels height. We assume the extra training stage of CLIP $336^2$ makes it better on the larger scale but worse on the smaller scale.

## 5.4 TRANSFERRED INSTRUCTION-FOLLOWING CAPABILITY

According to the dataset statistics (Table 1) and the visualization (Figure 11), our collected instruction-following data is not as diverse and substantial as LLaVA. This can be attributed to the relatively limited information given GPT-4 compared to five different human-written captions used in LLaVA. The content of text-rich images is also less diverse than that of natural images. While using more complex in-context examples can definitely stimulate generating more complicated instruction-following examples, it can also multiply the cost. In Appendix Figure 9, we demonstrate the transferred instruction-following capability of LLaVA, potentially from both the LLaVA data and the Vicuna backbone. While the extra data we add mainly focuses on understanding the visible texts within images, LLaVAR manages to build its reasoning, writing, and elaboration skills based on the top of its text recognition capability in an end-to-end manner. This allows users to interact with various online content based on simple screenshots.

## 6 CONCLUSION

In this work, we enhance visual instruction-tuned models in terms of their capability to read texts in images. Using text-rich images from the LAION dataset, we collect 422K noisy instruction-following examples using OCR results only and 16K high-quality instruction-following data based on text-only GPT-4. These two sets of data are leveraged to augment the pretraining stage and fine-tuning stage of LLaVA accordingly. Our model, LLaVAR, demonstrates superior performance in understanding texts within images and following human instructions on both prior benchmarks and real-world online content. Moreover, our analysis shows that the same augmented data is more effective with higher resolution. Additionally, using noisy instruction-following examples to augment pretraining essentially boosts the model performance without prompting GPT-4. For future work, we encourage exploration of **(i)** better image selection criteria or domain reweighting strategy (Xie et al., 2023) and **(ii)** more data-efficient and computation-efficient ways to augment instruction-following models with multimodal capability, especially in the high-res scenario.

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

| Instructions |
| --- |
| Identify any text visible in the image provided. |
| List all the text you can see in the given image. |
| Enumerate the words or sentences visible in the picture. |
| Describe any readable text present in the image. |
| Report any discernible text you see in the image. |
| Share any legible words or sentences visible in the picture. |
| Provide a list of texts observed in the provided image. |
| Note down any readable words or phrases shown in the photo. |
| Report on any text that can be clearly read in the image. |
| Mention any discernable and legible text present in the given picture. |

Table 6: Ten instructions asking for OCR results.

## A

**CLIP-based categorization**    Based on the observation of selected clusters, we divide the images used into 8 categories. For each category, we use one or multiple words as labels.

- **Quote & Meme**: "quote", "internet meme".
- **Poster**: "movie poster", "podcast poster", "TV show poster", "event poster", "poster",
- **Book Cover**: "book cover", "magazine cover".
- **Game Cover**: "game cover".
- **Ad & Product Packaging**: "ad", "advertisement", "food packaging", "product packaging".
- **Infographic**: "chart", "bar chart", "pie chart", "scatter plot".
- **Educational Material**: "exam paper", "quiz", "certificate", "book page".
- **Logo**: "logo".

For each word, we use the following templates to achieve embedding-space ensembling (Radford et al., 2021):

- "a photo of a {}."
- "a blurry photo of a {}."
- "a black and white photo of a {}."
- "a low contrast photo of a {}."
- "a high contrast photo of a {}."
- "a bad photo of a {}."
- "a good photo of a {}."
- "a photo of a small {}."
- "a photo of a big {}."

For each image, we calculate the similarity between the image and all words mentioned above using `CLIP-ViT-L/14`. If the highest similarity is less than $0.15$, we then classify the image into **Other**, otherwise we classify into the "super class" (e.g., **Poster**) of the word (e.g., "movie poster") with the highest similarity.

## B

**System Message**    adapted from Liu et al. (2023a).

*You are an AI visual assistant, and you are seeing a single image. What you see is provided with two OCR results and one image caption describing the information within the same image you are looking at. Image captions might include hallucinations, while OCR results are more accurate. Answer all questions with definite answers as you are seeing the image.*

*Design a conversation between you and a person asking about this photo. The answers should be in a tone that a visual AI assistant is seeing the image and answering the question. Ask diverse questions and give corresponding answers.*

*Include questions asking about the visual content of the image (e.g., the man, the sunset, the ocean.) and the texts contained in the image. Only include questions that have definite answers:*

*(1) one can see the content in the image that the question asks about and can answer confidently;*

*(2) one can determine confidently from the image that it is not in the image. Do not ask any questions that cannot be answered confidently;*

*(3) DO NOT mention OCR or image caption in your questions and answers;*

*(4) DO NOT ask about information from captions while it looks unrelated to or contradicts OCR results.*

*Also include complex questions that are relevant to the content in the image, for example, asking about background knowledge of the texts in the image, asking to discuss about the design of the image, etc. Again, do not ask about uncertain details. Provide detailed answers when answering complex questions. For example, give detailed examples or reasoning steps to make the content more convincing and well-organized. You can include multiple paragraphs if necessary.*

**Few-shot Examples**

*Human:*
*WHAT F YOUR BLESSINGS COME THROUGH RAINDROPS PDf Induded LAURAS-TOR Y DEVOHONAL BASED ON THE GRAAAMY VARD WIN ISONG mAT HAS IN-SEIRED MILLION*
*WHATIFYOUR BLESSINGS COMETHROUGHRAINDROPS PDF Included LAURASTORY A DEVOTIONAL BASEDON THE GRAMMY AWARD WINNI PISONGTHATHASINSPIREDMILLIONSE*
*a girl is standing in a field with a rainbow*
*GPT:*
*Question: What is the name of the devotional mentioned in the image?*
*Answer: The devotional is called "What If Your Blessings Come Through Raindrops" by Laura Story.*
*Question: What is special about the song mentioned in the image?*
*Answer: The song is special because it's a Grammy Award-winning song that has inspired millions of people."*

*Human:*
*One of the hardest things in life to accept is a called third strike Robert Frost te*
*One of the hardest things in life to accept is a called third strike Robert Frost quotefancy*
*a close up of a baseball glove*
*GPT:*
*Question: Why is the third strike hard to accept? Explain the quote to me.*
*Answer: The quote Öne of the hardest things in life to accept is a called third strikeättributed to Robert Frost is a metaphor for life situations. In baseball, a called third strike is when the umpire determines that a pitch was a strike, but the batter did not swing. This can be frustrating for the batter because they missed an opportunity to hit the ball due to either a lack of confidence or misjudgment.*
*In life, this metaphor refers to difficult moments where we might miss opportunities or face setbacks because of our own inaction or hesitation. The quote suggests that it is*

> *hard to accept these missed opportunities or challenges, especially when they result from our own choices or lack of action.*

| Method | Subject | | | Context Modality | | | Grade | | Average |
|---|---|---|---|---|---|---|---|---|---|
| | NAT | SOC | LAN | TXT | IMG | NO | G1-6 | G7-12 | |
| Human (2022) | 90.23 | 84.97 | 87.48 | 89.60 | 87.50 | 88.10 | 91.59 | 82.42 | 88.40 |
| GPT-3.5 (2022) | 74.64 | 69.74 | 76.00 | 74.44 | 67.28 | 77.42 | 76.80 | 68.89 | 73.97 |
| GPT-3.5 w/ CoT (2022) | 75.44 | 70.87 | 78.09 | 74.68 | 67.43 | 79.93 | 78.23 | 69.68 | 75.17 |
| LLaMA-Adapter (2023c) | 84.37 | 88.30 | 84.36 | 83.72 | 80.32 | 86.90 | 85.83 | 84.05 | 85.19 |
| MM-CoT$_{Base}$ (2023d) | 87.52 | 77.17 | 85.82 | 87.88 | 82.90 | 86.83 | 84.65 | 85.37 | 84.91 |
| MM-CoT$_{Large}$ (2023d) | 95.91 | 82.00 | 90.82 | 95.26 | 88.80 | 92.89 | 92.44 | 90.31 | 91.68 |
| LLaVA (2023a) | 90.36 | 95.95 | 88.00 | 89.49 | 88.00 | 90.66 | 90.93 | 90.90 | 90.92 |
| LLaVA+GPT-4 (2023a) | 91.56 | 96.74 | 91.09 | 90.62 | 88.99 | 93.52 | 92.73 | 92.16 | 92.53 |
| Chameleon (GPT-4) (2023) | 89.83 | 74.13 | 89.82 | 88.27 | 77.64 | 92.13 | 88.03 | 83.72 | 86.54 |
| LLaVAR | 91.79 | 93.81 | 88.73 | 90.57 | 88.70 | 91.57 | 91.30 | 91.63 | 91.42 |

Table 7: Results (accuracy %) on Science QA dataset. All baseline results are from Liu et al. (2023a); Lu et al. (2023). The categories are denoted as NAT: natural science, SOC: social science, LAN: language science, TXT: text context, IMG: image context, NO: no context, G1-6: grades 1-6, G7-12: grades 7-12.

# C

Details of evaluation datasets used in the main paper:

- ST-VQA (Furkan Biten et al., 2019) contains 31791 questions that require understanding the scene text, based on images from COCO (Lin et al., 2015), Visual Genome (Krishna et al., 2016), ImageNet (Deng et al., 2009), etc.

- TextVQA (Singh et al., 2019) contains 45,336 questions that need reading and reasoning about the text in images to answer, based on images from OpenImages (Krasin et al., 2017).

- OCR-VQA (Mishra et al., 2019) contains more than 1 million questions asking about information from book cover images (Iwana et al., 2016).

- DocVQA (Mathew et al., 2020) contains 50000 questions based on document images.

Details of extra datasets in Appendix:

- CT80 (Risnumawan et al., 2014) contains 80 images for curved text OCR evaluation. The formats of questions are: (1) "What is written in the image?" for English words. (2) "What is the number in the image?" for digit string.

- POIE (Singh et al., 2019) contains 3000 camera images collected from the Nutrition Facts label of products, together with 111,155 text instances. The format of questions is "What is {entity name} in the image?".

- ChartQA (Masry et al., 2022) includes 4,804 charts with 9608 human-written questions.

# D

**Results of other metrics**  The metric used for text-based VQA in the main paper is the standard practice in VQA benchmarks (Antol et al., 2015). For STVQA and DocVQA, previous works use ANLS (Average Normalized Levenshtein Similarity) as the metric (Furkan Biten et al., 2019; Mathew et al., 2020), which calculates the average normalized edit distance and only works for supervised models trained to output short and precise answers. It works badly for instruction-following models that usually output long sequences instead of brief answers. For reference, we provide more text-matching metrics (METEOR, Banerjee & Lavie, 2005, ROUGE-L, Lin, 2004, CIDEr, Vedantam et al., 2014) to demonstrate the improvement of our model (Table 8, 9, 10, 11), which works well

|  | Res. | METEOR | ROUGE-L | CIDEr |
|---|---|---|---|---|
| LLaVA | $224^2$ | 7.0 | 8.2 | 15.3 |
| LLaVAR |  | 10.0 | 11.4 | 24.5 |
| LLaVA | $336^2$ | 8.4 | 9.9 | 19.1 |
| LLaVAR |  | 12.8 | 14.3 | 30.9 |

Table 8: Results on ST-VQA using text-matching metrics.

|  | Res. | METEOR | ROUGE-L | CIDEr |
|---|---|---|---|---|
| LLaVA | $224^2$ | 8.7 | 10.5 | 12.2 |
| LLaVAR |  | 12.5 | 14.9 | 21.4 |
| LLaVA | $336^2$ | 9.9 | 12.1 | 15.3 |
| LLaVAR |  | 14.8 | 17.4 | 27.0 |

Table 9: Results on textVQA using text-matching metrics.

except for OCR-VQA. We assume these metrics are not valuable for OCR-VQA since the ground truth answers are usually too short.

# E

**Results on extra datasets** In Table 12, we provide results on three extra datasets: CT80 (OCR, Risnumawan et al., 2014), POIE (Information Extraction, Kuang et al., 2023), and ChartQA (Masry et al., 2022). We use the same VQA metric as other text-based VQA datasets. We observe similar trends as the main paper results: LLaVAR data significantly improves over the LLaVA baseline, usually more considerably in a higher resolution.

# F

**Comparison with mPLUG-Owl** We find that LLaVAR usually performs similarly well with mPLUG-Owl in the same $224^2$ resolution. We further clarify the setting differences between mPLUG-Owl and ours: mPLUG-Owl is trained on 1000M+ text-image pairs, while the original LLaVA is trained on about 0.6M text-image pairs. Our model, LLaVAR, is trained on about 1M text-image pairs. Within the same resolution, LLaVAR demonstrates a good performance with decent data efficiency.

We presume that training on large-scale non-OCR data improves OCR performance, as many of the captions in LAION datasets are equivalent to incomplete OCR results (Texts in an online image will sometimes appear in the captions). In the scale of our experiment, we observe similar improvement that just training on captions of text-rich images can help with text recognition capability: In Table 3, variant (4) is better than variant (1). However, training on captions only (variant (4)) is not as good as training on OCR-based data (variant (2)(6)), at least in the scale of our experiments.

**Results of finetuning mPLUG-Owl** To further validate the effectiveness of our collected data, we provide the results of finetuning mPLUG-Owl using our 16K GPT-4-based instruction-following data in Table 13. Though the mPLUG-Owl checkpoint is extensively trained on 1000M+ text-image pairs, we find that our data can boost performance in most cases, demonstrating the effectiveness of our data.

# G

**ScienceQA Results** Starting from our pretrained LLaVAR ($336^2$-based, without finetuning), we also report the results of further finetuning on the ScienceQA dataset (Lu et al., 2022) in Table 7, which is a multimodal multi-choice QA dataset covering diverse domains. Our motivation is that

|  | Res. | METEOR | ROUGE-L | CIDEr |
|---|---|---|---|---|
| LLaVA | $224^2$ | 0.2 | 0.1 | 0.0 |
| LLaVAR |  | 0.3 | 0.1 | 0.0 |
| LLaVA | $336^2$ | 0.3 | 0.1 | 0.0 |
| LLaVAR |  | 0.2 | 0.1 | 0.0 |

Table 10: Results on OCR-VQA using text-matching metrics.

|  | Res. | METEOR | ROUGE-L | CIDEr |
|---|---|---|---|---|
| LLaVA | $224^2$ | 3.8 | 4.8 | 6.3 |
| LLaVAR |  | 5.6 | 6.9 | 12.7 |
| LLaVA | $336^2$ | 4.6 | 5.6 | 8.7 |
| LLaVAR |  | 8.6 | 10.0 | 21.5 |

Table 11: Results on DocVQA using text-matching metrics.

some images in this dataset contain text descriptions and tables that require textual understanding within images. The LLaVAR model finetuned on ScienceQA achieves an average accuracy of 91.42%, better than LLaVA (90.92%), while the most considerable improvement comes from natural science questions (+1.43%).

# H

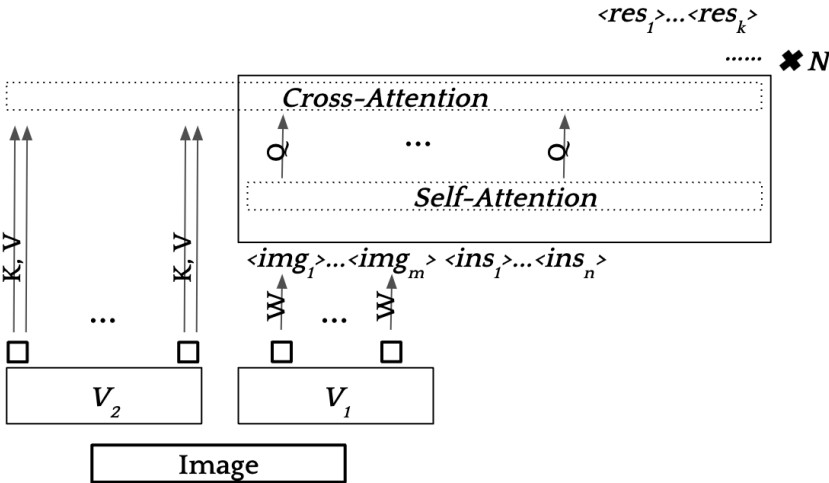

Figure 7: Illustration of the dual visual encoder system. Given an image, it is simultaneously processed by visual encoders $V_1$ and $V_2$. $V_1$ features are transformed by transformation matrix $W$ and directly used as input embeddings to the language model. For $V_2$ features, they are transformed by transformation matrix $K$ and $V$ and used as keys and values to calculate the cross attention in every transformer layer (assume there are $N$ layers), which uses the transformed hidden states (through $Q$) from the self-attention module as queries. For the language decoder $D$, the input is image tokens ($$) and instruction tokens ($$), while the target is response tokens ($<res>$).

The original version of LLaVAR only supports up to $336^2$ resolution, while our case study has also shown the threshold for the recognizable font size. Both suggest the difficulty of processing real-world high-res images without scaling and cutting. To this end, we test a dual visual encoder system for the high-res variant of LLaVAR, where a high-res visual encoder is added to work with the standard one. Ideally, the standard visual encoder extracts general, high-level information, while the high-res one specifically helps with detailed information.

|  | Res. | CT80 | POIE | ChartQA |
|---|---|---|---|---|
| BLIP-2 (2023c) † |  | 80.9 | 2.5 | 7.2 |
| OpenFlamingo (2023) † |  | 67.7 | 2.1 | 9.1 |
| MiniGPT4 (2023) † | $224^2$ | 57.3 | 1.3 | 4.3 |
| LLaVA (2023a) † |  | 61.1 | 2.1 | 7.3 |
| mPLUG-Owl (2023) † |  | 81.9 | 3.3 | 9.5 |
| LLaVA ‡ | $224^2$ | 61.5 | 1.9 | 9.2 |
| LLaVAR |  | 81.6 (+20.1) | 5.7(+3.8) | 10.2 (+1.0) |
| LLaVA ‡ | $336^2$ | 64.9 | 2.5 | 10.2 |
| LLaVAR |  | 83.0 (+18.1) | 8.7(+6.2) | 13.5 (+3.3) |

Table 12: Results (accuracy %) on three extra datasets: OCR, Information Extraction, and Chart Question Answering. We use † to refer to the results obtained from Liu et al. (2023c) and ‡ to refer to our reproduced results.

|  | ST-VQA | OCR-VQA | TextVQA | DocVQA | CT80 | POIE | ChartQA |
|---|---|---|---|---|---|---|---|
| mPLUG-Owl | 29.3 | 28.6 | 40.3 | 6.9 | 81.9 | 3.3 | 9.5 |
| mPLUG-Owl$_{ours}$ | 29.6 | 31.2 | 40.8 | 7.0 | 84.7 | 3.7 | 10.2 |

Table 13: Results (accuracy %) of finetuning mPLUG-Owl. mPLUG-Owl$_{ours}$ denotes mPLUG-Owl finetuned on our 16K GPT-4-based instruction-following data.

**Architecture** A high-res visual encoder usually outputs thousands of visual features. Simply following LLaVA to feed the transformed visual features into the context of the language decoder is impractical, as the maximum sequence length of the language decoder is usually 2048/4096. To this end, we propose to handle high-res visual features by cross-attention module and standard visual features by feature transformation. We depict the proposed system in Figure 7.

Specifically, given a standard visual encoder $V_1$, the extracted features are transformed into the word embedding space of the language decoder through a trainable projection matrix $W$. These transformed features are then concatenated with the word embeddings to build the input embeddings of the language decoder $D$.

$$\text{emb}(\langle \text{img}_1 \rangle), \cdots, \text{emb}(\langle \text{img}_m \rangle) = WV_1(I)$$
$$\text{input\_emb} = \textbf{concat}([\text{emb}(\langle \text{img}_1 \rangle), \cdots, \text{emb}(\langle \text{img}_m \rangle), \text{emb}(\langle \text{ins}_1 \rangle), \cdots, \text{emb}(\langle \text{ins}_n \rangle)]) \quad (1)$$

where $I$ is the input image, $V_1$ denotes extracting the grid features before the last transformer layer.

At the same time, we use the high-res visual encoder $V_2$ to extract high-res visual features, which are then transformed into keys/values as the inputs of the cross-attention module in transformer layers. Given $h^j$ as the hidden state before the cross-attention module in layer $j$,

$$\text{CrossAttention}(h, V_2, I) = \text{softmax}(\frac{Q^j h^j (K^j V_2(I))^T}{\sqrt{d}})V^j V_2(I) \quad (2)$$

where $Q^j, K^j, V^j$ denotes the query/key/value projection matrix in the $j$-th transformers layer. In practice, there is a pre-attention LayerNorm before calculating the attention and another output projection matrix $O^j$ to project the aggregated values back to the hidden space.

As the pretrained language decoder $D$ might only have self-attention modules, we manually add another cross-attention module after the original self-attention module in every transformer layer. Considering the random initialization of cross-attention modules might hurt the original language generation capability, we initialize the value projection matrix $V^j$ as a zero matrix and the output projection matrix $O^j$ as an identity matrix.

**Implementation** We use `CLIP-ViT-L/14` as the standard visual encoder. For the high-resolution encoder, we test two models: **(i)** `Pix2Struct-base` (Lee et al., 2022) is a visual

encoder trained on screenshot to HTML transformation. It supports up to 2048 patches with size $16^2$, equivalent to $1024 * 512$. **(ii)** `ConcatCLIP` refers to using 16 `CLIP-ViT-L/14` models to encode the $4 * 4$ grids of images separately and then concatenate the extracted features together. In other words, it supports $896^2$ resolution. We use Vicuna-7B as the language decoder for the high-res version of LLaVAR.

**Training**    Only cross-attention modules and the projection matrix $W$ are trained during pretraining, while visual encoders and the language decoder are frozen. Cross-attention modules, the projection matrix $W$, and the language decoder $D$ are trained during finetuning.

**Data**    To fully unlock the potential of the augmented visual encoder, we also double the number of pretraining examples using the same criteria mentioned in Section 3. This corresponds to the variant (g) in Table 4.

|  | ST-VQA | OCR-VQA | TextVQA | DocVQA |
|---|---|---|---|---|
| `Pix2Struct` + LLaVA | 21.9 | 11.8 | 28.7 | 4.4 |
| `Pix2Struct` + LLaVAR | 35.8 (+13.9) | 30.7 (+18.9) | 45.6 (+16.9) | 15.3 (+10.9) |
| `ConcatCLIP` + LLaVA | 23.1 | 14.2 | 30.5 | 5.1 |
| `ConcatCLIP` + LLaVAR | 42.1 (+19.0) | 30.8 (+16.8) | 52.1 (+21.6) | 18.5 (+13.4) |

Table 14: Additional results on the dual visual encoder system.

**Discussion**    We report the performance of augmented architecture, using either LLaVA or LLaVAR data in Table 14. By comparing the relative improvement in Table 2 and 14, we find that higher-resolution models benefit more from our collected data, suggesting our data is underutilized in the original LLaVA architecture.

## I

|  | Res. | Correct % | Partially Correct% |
|---|---|---|---|
| LLaVA | $224^2$ | 1.6% | 8.7% |
| LLaVAR |  | 6.8% | 22.8% |
| LLaVA | $336^2$ | 2.2% | 11.2% |
| LLaVAR |  | 9.0% | 26.8% |

Table 15: Statistics of correct answers and partially correct answers on OCR-VQA.

**The impact of OCR spelling errors**    We study such OCR errors by studying 1673 examples from OCR-VQA, which have ground truth answers with more than ten characters. We (i) define "correct" as the ground truth answers that are exactly in the predictions, and (ii) define "partially correct" as there exists a substring in the prediction that has high enough similarity with the ground truth but not the same. Specifically, we look at all substrings with the same length of the ground truth in the prediction to calculate ANLS (Average Normalized Levenshtein Similarity) and regard the prediction as "partially correct" if the highest ANLS is greater or equal to 0.5 but smaller than 1.

We find that many predictions can be considered partially correct, indicating the actual performance of tested models is better than the reported accuracy numbers. However, the percentage of partially correct predictions is highly correlated with the percentage of correct predictions. Therefore, we believe that the current metrics can effectively compare the performance of different models.

## J

**Ablation Study on Instruction-following Evaluation**    To provide a more detailed understanding of LLaVAR's data, we provide the ablation study results on 90 instruction-following questions from

|  | Conversation | Detail | Complex |
|---|---|---|---|
| LLaVA | 83.6 | 78.1 | 95.2 |
| LLaVA + $R_{pretrain}$ | 86.7 | 79.3 | 95.1 |
| LLaVA + $R_{finetune}$ | 79.0 | 79.4 | 98.0 |
| LLaVAR | 84.4 | 78.9 | 96.6 |

Table 16: Ablation Study on GPT-4-based Instruction-following Evaluation from LLaVA (Liu et al., 2023a).

| | Question: Based on the title and the image on the cover, what can be inferred about the content of "Bo's Lasting Lessons" and its potential target audience? | Relative Score |
|---|---|---|
| GPT-4 (Text-Only) | From the title and image, it can be inferred that "Bo's Lasting Lessons" is a book about leadership lessons taught by a legendary coach, likely in the context of sports. The image of men running suggests that the book might be focused on a specific sport, such as football or track and field. The target audience could be individuals interested in coaching, sports, or leadership in general, looking to learn from the experiences and teachings of a successful coach. | - |
| LLaVA | The book "Bo's Lasting Lessons: The Timeless Teachings of Bo Schembechler" is written by Bo Schembechler and Jim Harshaw. Given the title and the image on the cover, which shows a football coach running across the field with his players in pursuit, it can be inferred that the content of the book is likely focused on football and the strategies, techniques, or principles that can help players improve their performance and achieve success in the sport. The potential target audience for this book could be football players, coaches, and fans who are interested in learning more about the game and the experiences and insights of a seasoned coach like Bo Schembechler. | 75% |
| LLaVAR | "Bo's Lasting Lessons" appears to be a book that shares the legacy coach Bo Schembechler's teachings on leadership and timeless fundamentals. The image on the cover, featuring a football coach leading his team, further supports this inference. The book might be targeted towards sports enthusiasts, particularly fans of football, and people interested in learning about leadership principles and fundamentals from a renowned coach's perspective. The audiobook, read by John H. Mayer and featuring a foreword and conversation with John U. Bacon, suggests that it could also appeal to audiences who prefer listening to books or want to gain more insights from the author and other experts. | 129% |

Table 17: An example for our collected evaluation data, corresponding to the "**Read**" dimension of GPT-4-based instruction-following evaluation. The corresponding image is Figure 8 in Appendix.

LLaVA (Liu et al., 2023a). We find that including pretraining data improves the conversation capability, probably because longer training data leads to generating longer responses (Table 1), which is favored by GPT-4 (Xu et al., 2023). On the other hand, including only finetuning data hurts the conversation capability but increases complex reasoning. Combining pretraining and finetuning data improves the trade-off between conversation and complex reasoning. Such results prove that incorporating our data will at least not harm the performance of interacting with natural images.

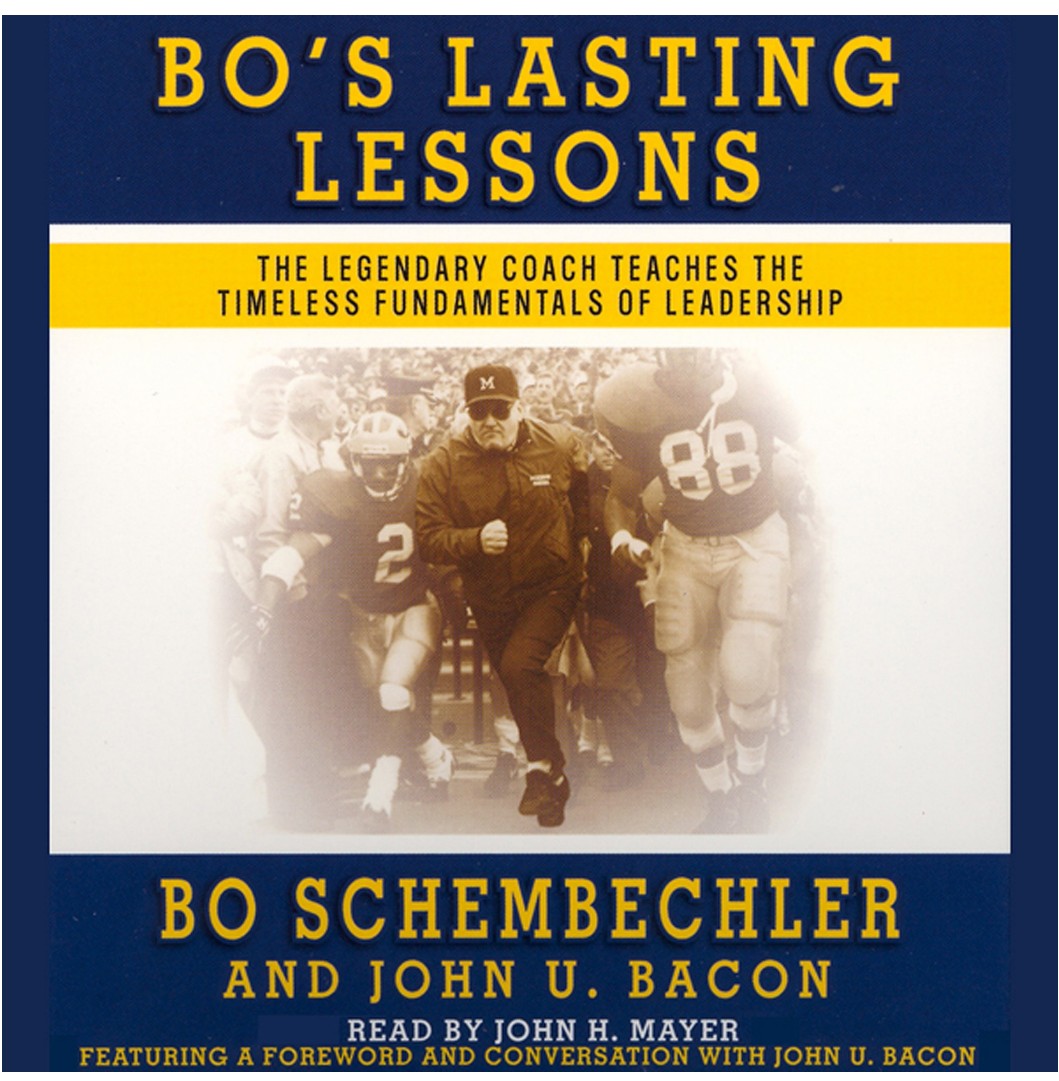

Figure 8: An example for the Read dimension of GPT-4-based instruction-following evaluation.

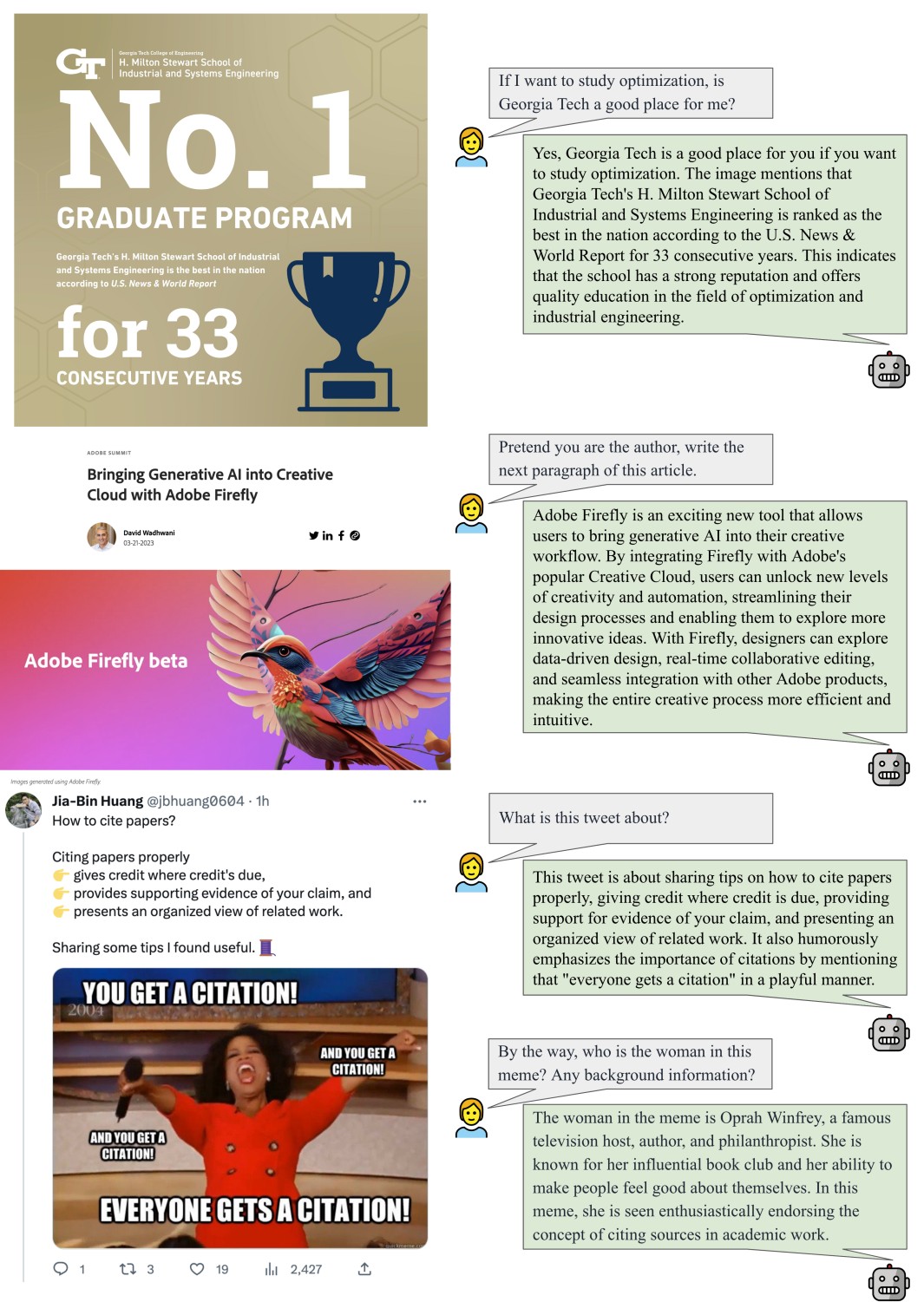

Figure 9: Transferred instruction-following capability of LLaVAR.

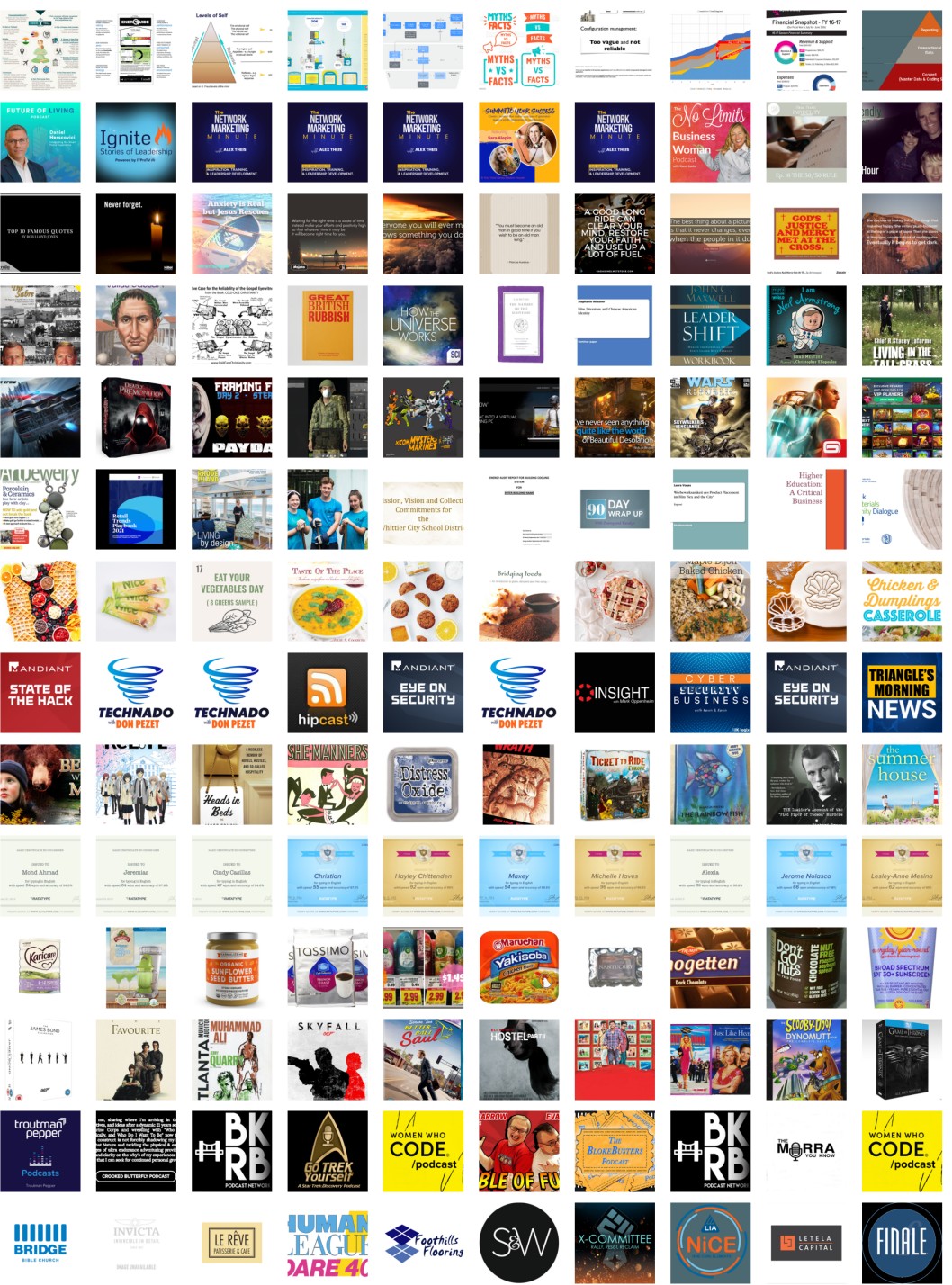

Figure 10: All 14 clusters we selected as text-rich images. Each row corresponds to one cluster, where we show ten randomly sampled examples before de-duplication.

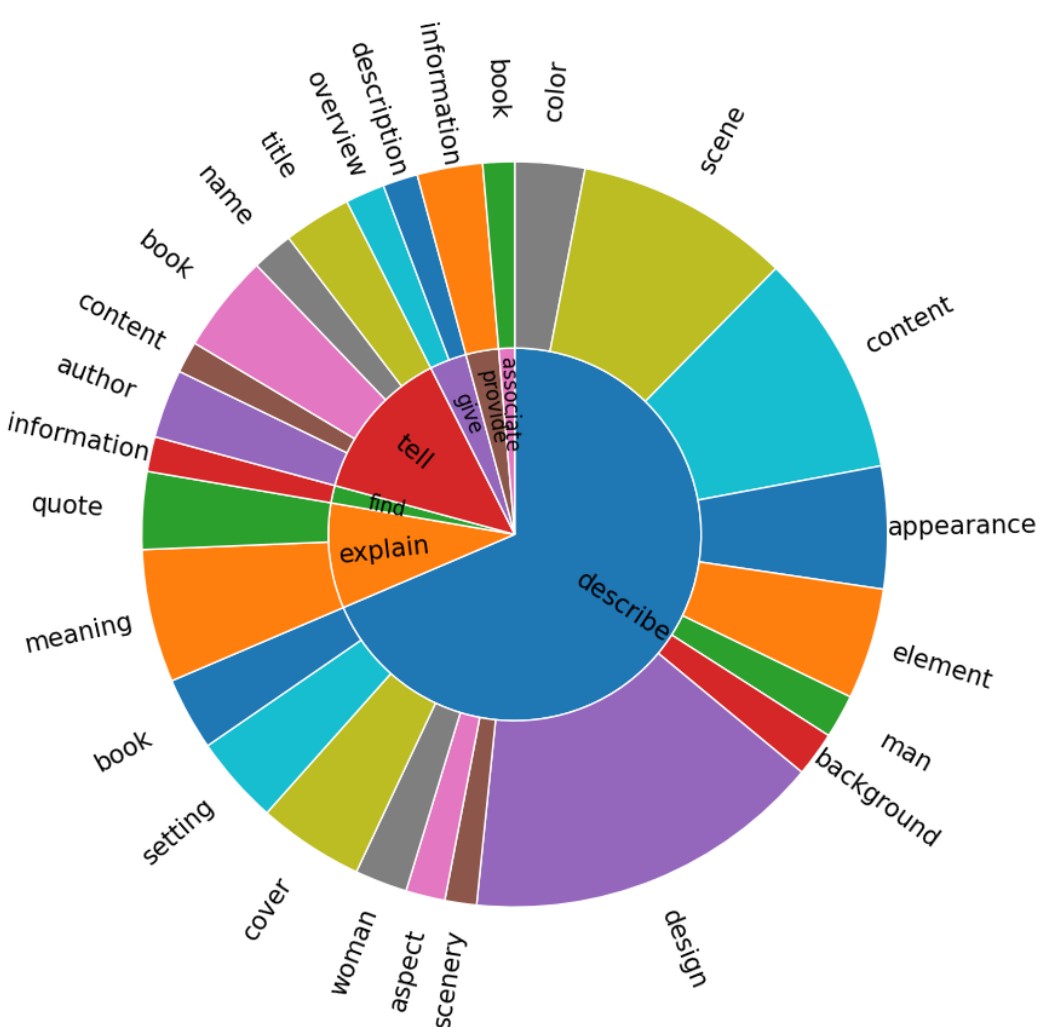

Figure 11: Visualization of collected instructions.

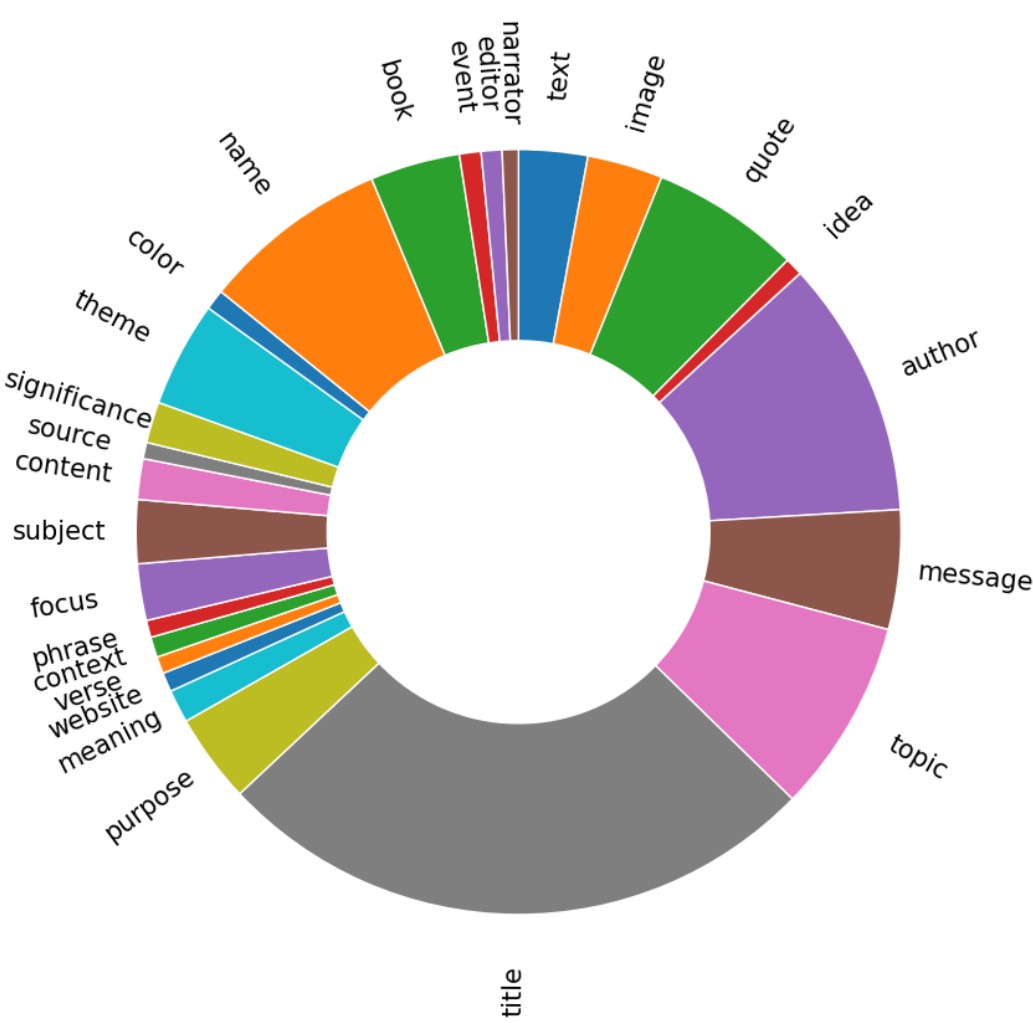

Figure 12: Visualization of collected instructions.

