# OpenReview forum: "Enhanced Visual Instruction Tuning for Text-Rich Image Understanding"
_ICLR.cc/2024/Conference — Submitted to ICLR 2024_

### Official Review · Reviewer_bdXG · 2023-10-31

**Soundness:** 3 good
**Presentation:** 3 good
**Contribution:** 2 fair
**Rating:** 5
**Confidence:** 4

**Summary:**

This work improves the collection pipeline of instruction-following data, allowing for the collection of a large-scale dataset of text-rich images. Leveraging GPT-4, this work further constructs an instruction tuning dataset consisting of 422K noisy pretraining data and 16K conversations and validates it on the recent work LLAVA. The results on multiple text-based VQA datasets show that this dataset improves the performance of LLAVA in text understanding. Case analysis also demonstrates that LLAVAR has stronger image-text understanding abilities than LLAVA.

**Strengths:**

1. The paper is well organized and easy to follow.
2. The improved data collection pipeline overcomes the limitations of the existing dataset, which lacks text-rich images and relevant instruction-tuning annotations.
3. The explanation of the data collection process is detailed, and the relevant experimental analysis provides support for investigating how to enhance visual Large Language Models' understanding of text-rich visual content.

**Weaknesses:**

1. The contribution of this paper appears to be limited. The proposed data collection pipeline in this paper is based on the one used in LLAVA, but with incremental improvements. Similarly, the model architecture in this paper also follows LLAVA without a specific design for text-rich scenarios.
2. The dataset introduced in this paper only brings limited improvement. In Table 2, LLAVAR achieves comparable performance with mPLUG-Owl, a model that was not trained on a text-rich dataset. The paper also does not provide a detailed comparison with other state-of-the-art models in the field of text-rich image understanding, which would help to better understand the relative performance of the proposed method.
3. The fragmented OCR results in a few words may also exist in real-world text-rich data such as poster, table, directly removing this kind of data may be also different from the real-world distribution.

**Questions:**

1. How is the performance if your dataset is trained based on mPLUG-Owl? Since it is a high baseline of your method. Can you further fine-tune other open-source models using the dataset from this article and provide performance comparisons?
2. Can you provide experimental results and analysis on more text-rich image understanding  benchmarks (e.g., Information Extraction, Document Classification, OCR).

---

> ### Author Response · Authors · 2023-11-20
>
> Thank you for acknowledging the organization and clarity of our paper, as well as appreciating the advancements we've made in data collection and our detailed analysis.
>
> For __comparison with baselines/methodology novelty/evaluation scenarios__, please refer to the general responses.
>
> __LLAVAR achieves comparable performance with mPLUG-Owl, a model that was not trained on a text-rich dataset? :__
>
> It is hard to conclude that mPLUG-Owl is a model not trained on a text-rich dataset, as many of the captions in LAION datasets are equivalent to incomplete OCR results (Texts in an online image will sometimes appear in the captions). In the scale of our experiment, we observe similar improvement that just training on captions of text-rich images can help with text recognition capability (In Table 3, variant (4) is better than variant (1)). However, training on captions only (variant (4)) is not as good as training on OCR-based data (variant (2)(6)), at least in the scale of our experiments. We assume training on captions can be powerful enough for the scale of mPLUG-Owl (1000M+ text-image pairs). However, we believe our data is lightweight and effective, and our pipeline is more customizable.
>
> __On fragmented OCR results:__
>
> The reason why we tried to remove fragmented OCR results from prompting GPT-4 is to remove repetitive and meaningless questions generated by GPT-4. If there are only a few unrelated words presented to GPT-4, it is hard to generate meaningful question-answer pairs for the texts. Note that (1) our noisy instruction-following data based on raw OCR results still contain such fragmented OCR results. (2) Our experiments show that the learned text recognition capability transfers well to scenarios like ST-VQA and textVQA, where the texts in images are usually fragmented words.

---

### Official Review · Reviewer_miv5 · 2023-10-31

**Soundness:** 3 good
**Presentation:** 3 good
**Contribution:** 2 fair
**Rating:** 6
**Confidence:** 3

**Summary:**

The paper collects noisy and high-quality instruction-following data to enhance visual instruction tuning for text understanding in images. Their model LLaVAR incorporates this new data and improves performance on text VQA and instruction following for text-rich images. The enhanced capability allows more natural interaction based on real-world online content combining text and images.

**Strengths:**

1. The paper focuses on an important problem of improving OCR ability for multimodal LLMs like LLaVA.
2. It identifies that key factors to improve model performance are training data and image resolutions. To address this, the paper collects over 400k specialized training examples to enhance OCR capabilities.
3. Extensive experiments verify the effectiveness of the proposed training data.

**Weaknesses:**

1. The conclusions are a bit obvious - that higher resolution inputs and more specialized training data improve LLaVA's OCR performance.
2. The most important contribution of the paper is the collected dataset. It succeeds in showing the data improves LLaVA's OCR capabilities, but does not demonstrate it is superior to other visual instruction datasets. For example, mPLUG-Owl has comparable OCR performance to LLaVAR under the same resolution in Table 2. This raises the question of whether OCR-specific data is needed, or if the scale of data in the paper is insufficient.
3. The evaluation is limited, mostly relying on 4 OCR QA datasets. As the authors admit in Fig 4(5), this evaluation may be unreliable. More scenarios like the LLaVA benchmark would be expected, especially in ablation studies.

**Questions:**

1. Why did the authors collect data based on LAION, rather than some well-annotated OCR dataset?

---

> ### Author Response · Authors · 2023-11-20
>
> Thank you for recognizing the strengths of our approach and the efforts we put into enhancing OCR capabilities and data collection for multimodal language models.
>
> For __comparison with baselines/methodology novelty/Evaluation metric/Evaluation Scenarios__, please refer to the general response.
>
> __Whether OCR-specific data is needed:__
>
> Compared to hundreds of millions of text-image pairs used in mPLUG-owl, we acknowledge that the scale of our data is relatively limited, which is also relatively affordable for most academic labs. We presume that training on large-scale non-OCR data improves OCR performance, as many of the captions in LAION datasets are equivalent to incomplete OCR results (Texts in an online image will sometimes appear in the captions). In the scale of our experiment, we observe similar improvement that just training on captions of text-rich images can help with text recognition capability (In Table 3, variant (4) is better than variant (1)). However, training on captions only (variant (4)) is not as good as training on OCR-based data (variant (2)(6)), at least in the scale of our experiments. We assume such a training signal can be strong enough for the scale of mPLUG-Owl. In general, we believe OCR-specific data is necessary for data efficiency.
>
> __Ablation study on the LLaVA benchmark__
>
> |                        | conversation | detail | complex |
> |------------------------|--------------|--------|---------|
> | LLaVA                  | 83.6         | 78.1   | 95.2    |
> | LLaVA + $R_{pretrain}$ | 86.7         | 79.3   | 95.1    |
> | LLaVA + $R_{finetune}$ | 79.0         | 79.4   | 98.0    |
> | LLaVAR                 | 84.4         | 78.9   | 96.6    |
>
> We find that including pretraining data improves the conversation capability, probably because longer training data leads to generating longer responses (Table 1). On the other hand, including finetuning data only hurts the conversation capability but increases complex reasoning. Combining pretraining and finetuning data improves the trade-off between conversation and complex reasoning. Generally speaking, GPT-4-based evaluation is not very robust, as there are some clear clues that it favors long responses [1]. By providing results on the LLaVA benchmark, we prove that incorporating our data will at least not harm the performance of interacting with natural images.
>
> __Why based on LAION rather than well-annotated OCR datasets:__
>
> The well-annotated OCR datasets are usually restricted to certain domains such as black-and-white documents, book covers, etc. As shown in Appendix Figure 10, the text-rich images in the LAION dataset, which contains all kinds of text-image pairs from the internet, are diverse and usually interleaved with natural images. We believe our instruction-following dataset based on LAION suffers from a relatively small domain shift compared to previously collected instruction-following data based on COCO, thus beneficial for potential knowledge transfer (Section 5.4). Also, we believe that collecting data based on real-world documents and well-annotated OCR datasets is an important next step to extend the scope of the data.
>
> [1] Canwen Xu, Daya Guo, Nan Duan, and Julian McAuley. Baize: An open-source chat model with parameter-efficient tuning on self-chat data, 2023.

---

> ### Comment · Reviewer_miv5 · 2023-11-23
> **Thanks for the responses.**
>
> Thanks for the responses of the authors, which have solved most of my concerns.
>
> I would like to lift my score if the authors provide a revised paper with the new experiment results.
>
> Best.

---

> > ### Author Response · Authors · 2023-11-23
> > **Paper updated**
> >
> > Dear Reviewer miv5,
> >
> > We’ve updated the paper to include the new experiment results.
> >
> > Thank you for your feedback and effort in reviewing our paper.

---

> > > ### Comment · Reviewer_miv5 · 2023-11-23
> > > **Thanks for the updated paper**
> > >
> > > Thanks for the authors' responses.
> > >
> > > Most of my concerns about experiments have been solved. I have lifted the score.

---

### Official Review · Reviewer_2XNz · 2023-10-31

**Soundness:** 2 fair
**Presentation:** 2 fair
**Contribution:** 2 fair
**Rating:** 5
**Confidence:** 4

**Summary:**

This paper proposes a methodology to improve the text reading capability of large language and visual assistants. There are mainly two contributions: the data collection procedure and the improvement of LLaVA with the data. Text-rich images are collected by applying an image classifier with some filtering criteria. Off-the-shelf OCR tools are then used to obtain the texts in the images. A pretraining task is defined to output the transcribed texts as target. For finetuning, GPT-4 is used to generate instruction-following data. GPT-4 is asked to generate a sequence of question-answer pairs. The model is finetuned with the generated data. The experimental results confirm that LLaVAR improves LLaVA for tasks requiring reading texts. The code, data and model will be released to the public.

**Strengths:**

This paper shows a practical way to collect a large amount of text-rich data. The quality of the data is confirmed by the experiments where the training with the collected data improves the model. The data will be released to the public and the community will be able to benefit from the work.

The methodologies to generate the pretraining and finetuning data are reasonable. The use of GPT-4 to generate instruction-following data is very similar to the idea of LLaVA, but it seems also effective to generate such data for tasks requiring reading texts.

**Weaknesses:**

What seems important to improve text reading capability of this type of models is to train the models with a task that requires to read texts. This work also does it by generating data with OCR-ed texts and defining tasks that require reading texts. As expected, it improves the model in terms of text reading capability. However, the problem is that this seems a shared problem in this field and there are other studies that tried to improve text reading capabilities of this type of models (e.g. PreSTU, Pix2Struct). There is no discussion in this aspect and it looks like this is yet another attempt with the same objective. It would be required to make the novelty and advantage clear against other studies.

This is essentially an extension of LLaVA with OCR tasks. It is certainly important to improve text reading capability of this type of models, but it looks a little bit incremental in terms of methodological novelty.

**Questions:**

I wanted to understand the detail of "GPT-4-based instruction-following evaluation". My assumption was that GPT-4 was treated as Oracle (or GT) and some scores were computed against it. However, it was not very clear how text-based GPT-4 can be used to generate GT for tasks with image inputs. Also, how to compute the scores was not clear.

**Details Of Ethics Concerns:**

No concern.

---

> ### Author Response · Authors · 2023-11-20
>
> Thank you for recognizing the practicality and effectiveness of our data collection and training methodologies, as well as the potential community benefits from the release of our data and model.
>
> For __methodology novelty__, please take a look at the general responses.
>
> __Comparing to previous studies like PreSTU, Pix2Struct:__
>
> Our work focuses on improving the text recognition capability of the multimodal instruction-following model, which can be built on related prior work like PreSTU and Pix2Struct. Assuming we have a good frozen image encoder (which can be CLIP, Pix2Struct, or Pix2Struct), we study how to align its text recognition capability with large language models (pretraining stage) and how to maintain and acquire such capability during instruction-following (finetuning stage). In current Table 4 and Appendix E, we provide the results for using Pix2Struct to augment CLIP and demonstrate its improvement over CLIP $224^2$.
>
> We assume the amount of data needed for feature alignment should be much less than that needed for feature learning. As our data and pipeline focus on feature alignment, we believe it can naturally benefit from any advanced image encoders like Pix2Struct and PreSTU.
>
> __GPT-4-based instruction-following evaluation__
>
> GPT-4’s responses are treated as oracles. We provide text-only GPT-4 with detailed descriptions of the image (human-written captions, OCR results) and collect feedback as oracles on related questions. To calculate the score, we provide text-only GPT-4 with the detailed description again, together with one question and two answers (one from text-only GPT-4, one from the model we want to test), and ask GPT-4 to give scores to the two answers (1 ~ 10). The final score is the ratio between the average score of the tested model and the average score of GPT-4. For example, “83.1” in Table 5 means its score is 83.1% of GPT-4’s score.

---

> > ### Comment · Reviewer_2XNz · 2023-11-23
> > **Thank you for the response**
> >
> > Thank you for the response. I'm going to definitely reflect the rebuttal to make my final decision. Thanks again for your hard work.

---

### Official Review · Reviewer_sAP9 · 2023-11-01

**Soundness:** 3 good
**Presentation:** 3 good
**Contribution:** 3 good
**Rating:** 6
**Confidence:** 4

**Summary:**

This paper enhances the visual text understanding ability of large multimodal models by instruction tuning methods. First, two sets of noisy and high-quality instruction-following data are constructed. Specifically, the high-quality instruction-following data are generated by prompting text-only GPT-4 with OCR results and captions. Then, a two-stage training strategy is developed, with the first stage learning OCR capability and the second stage learning high-level understanding capability. Extensive experiments verify that the proposed LLaVAR model can improve performance on both natural and text-rich images.

**Strengths:**

1) This work a pioneering exploration of visual instruction tuning for text images, which can provide some useful insights to the community.
2) The proposed model has ability to deal with high-resolution images by integrating extra visual encoders and cross-attention modules.
3) The experiments are basically sufficient to demonstrate the superiority of the LLaVAR method, especially relative to the LLaVA baseline.

**Weaknesses:**

1) This work is less innovative in approach, as it mainly focuses on the construction of instruction-following data, while the proposed model and implementation pipeline basically follow LLaVA.
2) From the results in Table 2, LLaVAR has no significant performance advantage compared with existing methods under the same resolution (2242), such as mPLGU-Owl (2023). Besides, it is better to evaluate the parameter sizes of these comparison models.
3) In Section 5.3, only one case study is carried out, so the derived conclusion is hardly convincing.
4) In Figure 7, the notations are not clearly explained, and the implementation details cannot be visually reflected in this figure.

**Questions:**

1) As can be seen in Figure 4 and Figure 5, OCR errors are inevitable, e.g., “Boynton” vs. “Byington”. Can you provide some results to analyze the impact of OCR errors?
2) As mentioned in the paper, the adopted metric only considers the recall, so it is not very reliable. Have you tried other quantitative metrics to prove the effectiveness of the method, such as the metrics designed for the image captioning task?
3) What does “temperature” refer to in the first paragraph of Section 5?

---

> ### Author Response · Authors · 2023-11-20
>
> Thank you for recognizing our work's innovative aspects and potential impact!
> We address your comments below:
>
> For __comparison with baselines/methodology novelty/evaluation metric__, please refer to the general responses.
>
> __Table of language model parameter size and training data size.__
>
> |                  | Language Model Parameter | Training data size |
> |------------------|--------------------------|--------------------|
> | BLIP-2-FlanT5XXL | 11B                      | 129M               |
> | OpenFlamingo     | 7B                       | 15M                |
> | MiniGPT4         | 13B                      | 5.0M               |
> | LLaVA            | 13B                      | 0.6M               |
> | mPLUG-owl        | 7B                       | 1112M              |
> | LLaVAR           | 13B                      | 1.0M               |
>
> As all baseline models use ViT CLIP 224 as their visual encoder, we list the parameter size of their language model above, together with their training data size.
>
> In our experiments, we find that the language model size has a trivial effect (see the table below) on the performance related to text-based VQA. We believe data efficiency is a more important factor to consider while comparing performance.
>
> |            | ST-VQA | OCR-VQA | TextVQA | DocVQA |
> |------------|--------|---------|---------|--------|
> | LLaVAR 7B  | 28.9   | 25.6    | 37.8    | 6.2    |
> | LLaVAR 13B | 30.2   | 23.4    | 39.5    | 6.2    |
>
> __More comprehensive case study:__
>
> The case study is now conducted on 825 examples from OCR-VQA. We have updated Figure 6 in the draft. In this large-scale case study, we still observe a threshold for recognizable texts for both $224^2$-based and $336^2$-based LLaVAR as the accuracy sharply decreases when the height is smaller than 7 pixels.
>
> __On Figure 7:__
>
> We have provided an updated Figure 7 with detailed captions. A sketch of the implementation details is as follows: Given an image, it is simultaneously processed by visual encoders $V_1$ and $V_2$. $V_1$ features are transformed by transformation matrix $W$ and directly used as input embeddings to the language model. For $V_2$ features, they are transformed by transformation matrix $K$ and $V$ and used as keys and values to calculate the cross attention in every transformer layer (assume there are $N$ layers), which uses the transformed hidden states (through $Q$) from the self-attention module as queries.
>
> __The impact of OCR errors:__
>
> We take 1673 examples from OCR-VQA, which have ground truth answers with more than 10 characters, to study such OCR errors. We (i) define “correct” as the ground truth answers that are exactly in the predictions, and (ii) define “partially correct” as there exists a substring in the prediction that has high enough similarity with the ground truth but not the same. Specifically, we look at all substrings with the same length of the ground truth in the prediction to calculate ANLS (Average Normalized Levenshtein Similarity) and regard the prediction as “partially correct” if the highest ANLS is greater or equal to 0.5 but smaller than 1.
>
> |           | Correct % | Partially Correct% |
> |-----------|-----------|--------------------|
> | LLaVA224  | 1.6%      | 8.7%               |
> | LLaVAR224 | 6.8%      | 22.8%              |
> | LLaVA336  | 2.2%      | 11.2%              |
> | LLaVAR336 | 9.0%      | 26.8%              |
>
> We find that a considerable amount of predictions can be considered partially correct, which indicates the actual performance of tested models is better than the reported accuracy numbers. However, the percentage of partially correct predictions is highly correlated with the percentage of correct predictions. Therefore, we believe that the current metrics can effectively compare the performance of different models.
>
> __Clarification on the term “temperature”:__
>
> The temperature used to sample examples from language models.

---

### Author Response · Authors · 2023-11-20
**General response (1)**

We thank the reviewers for their detailed feedback and insightful comments.

__Comparing to mPLUG-Owl within the same resolution:__

mPLUG-Owl is trained on 1000M+ text-image pairs, while the original LLaVA is trained on about 0.6M text-image pairs. Our model, LLaVAR, is trained on about 1M text-image pairs. Within the same resolution, LLaVAR demonstrates a good performance with decent data efficiency.

Furthermore, we provide the results of finetuning mPLUG-Owl using our 16K GPT-4-based Instruction-following Data.

|            | ST-VQA | OCR-VQA | TextVQA | DocVQA | CT80 | POIE | ChartQA |
|------------|--------|---------|---------|--------|------|------|---------|
| mPLUG      | 29.3   | 28.6    | 40.3    | 6.9    | 81.9 | 3.3  | 9.5     |
| mPLUG_ours | 29.6   | 31.2    | 40.8    | 7.0    | 84.7 | 3.7  | 10.2    |


(We added three extra datasets: CT80 [1] (OCR), POIE [2] (Information Extraction), and ChartQA [3].)

We find that our data can boost the performance in most cases, though the mPLUG-Owl baseline is extensively trained on 1000M+ text-image pairs.

__Methodology Novelty:__

Though the main part of our study follows the original architecture of LLaVA, we also extend the architecture by connecting the language model with another high-resolution encoder through cross-attention.

In the updated draft, we include the comparison between LLaVA and LLaVAR on those extended architectures.

|                     | ST-VQA       | OCR-VQA      | TextVQA      | DocVQA       |
|---------------------|--------------|--------------|--------------|--------------|
| Pix2Struct + LLaVA  | 21.9         | 11.8         | 28.7         | 4.4          |
| Pix2Struct + LLaVAR | 35.8 (+13.9) | 30.7 (+18.9) | 45.6 (+16.9) | 15.3 (+10.9) |
| ConcatCLIP + LLaVA  | 23.1         | 14.2         | 30.5         | 5.1          |
| ConcatCLIP + LLaVAR | 42.1 (+19.0) | 30.8 (+16.8) | 52.1 (+21.6) | 18.5 (+13.4) |

Note:

1. Pix2Struct is a visual encoder trained on screenshots to HTML transformation. It supports up to 2048 patches with size $16^2$, equivalent to $1024 * 512$.
2. ConcatCLIP refers to using 16 CLIP-ViT-L/14 models to encode the $4 * 4$ grids of images separately and then concatenate the extracted features together. In other words, it supports $896^2$ resolution.

Instead of pursuing the best performance within the low resolution (e.g., $224^2$), our study reveals that higher-resolution models benefit more from our collected data, suggesting our data is underutilized in the original LLaVA architecture. We hope our result can inspire future design to leverage existing data better with suitable visual encoders.

__Evaluation Scenarios:__

We added three extra datasets: CT80 [1] (OCR), POIE [2] (Information Extraction), and ChartQA [3]. We use the same VQA metric as other text-based VQA datasets.

|              | CT80 | POIE | ChartQA |
|--------------|------|------|---------|
| BLIP-2       | 80.9 | 2.5  | 7.2     |
| OpenFlamingo | 67.7 | 2.1  | 9.1     |
| MiniGPT4     | 57.3 | 1.3  | 4.3     |
| mPLUG-Owl    | 81.9 | 3.3  | 9.5     |
| LLaVA224     | 61.5 | 1.9  | 9.2     |
| LLaVAR224    | 81.6 | 5.7  | 10.2    |
| LLaVA336     | 64.9 | 2.5  | 10.2    |
| LLaVAR336    | 83.0 | 8.7  | 13.5    |

[1]  Anhar Risnumawan, Palaiahnakote Shivakumara, Chee Seng Chan, and Chew Lim Tan. A robust arbitrary text detection system for natural scene images. Expert Syst. Appl., 41:8027–8048, 2014.

[2] Jianfeng Kuang, Wei Hua, Dingkang Liang, Mingkun Yang, Deqiang Jiang, Bo Ren, Yu Zhou, and Xiang Bai. Visual information extraction in the wild: Practical dataset and end-to-end solution. arXiv preprint arXiv:2305.07498, 2023.

[3] Ahmed Masry, Do Xuan Long, Jia Qing Tan, Shafiq Joty, and Enamul Hoque. Chartqa: A benchmark for question answering about charts with visual and logical reasoning. arXiv preprint arXiv:2203.10244, 2022.

---

### Author Response · Authors · 2023-11-20
**General response (2)**

We thank the reviewers for their detailed feedback and insightful comments.

__Evaluation Metric:__

The metric used for text-based VQA is the standard practice in VQA benchmarks. For STVQA and DocVQA, previous works use ANLS (Average Normalized Levenshtein Similarity) as the metric, which calculates the average normalized edit distance, which only works for supervised models that are trained to output short and precise answers. It works badly for instruction-following models that usually output long sequences instead of brief answers. For reference, we provide more text-matching metrics to demonstrate the improvement of our model, which works well except for OCR-VQA.

ST-VQA:

|           | METEOR | ROUGE-L | CIDEr |
|-----------|--------|---------|-------|
| LLaVA224  | 7.0    | 8.2     | 15.3  |
| LLaVAR224 | 10.0   | 11.4    | 24.5  |
| LLaVA336  | 8.4    | 9.9     | 19.1  |
| LLaVAR336 | 12.8   | 14.3    | 30.9  |

textVQA:

|           | METEOR | ROUGE-L | CIDEr |
|-----------|--------|---------|-------|
| LLaVA224  | 8.7    | 10.5    | 12.2  |
| LLaVAR224 | 12.5   | 14.9    | 21.4  |
| LLaVA336  | 9.9    | 12.1    | 15.3  |
| LLaVAR336 | 14.8   | 17.4    | 27.0  |

OCR-VQA:

|           | METEOR | ROUGE-L | CIDEr |
|-----------|--------|---------|-------|
| LLaVA224  | 0.2    | 0.1     | 0.0   |
| LLaVAR224 | 0.3    | 0.1     | 0.0   |
| LLaVA336  | 0.3    | 0.1     | 0.0   |
| LLaVAR336 | 0.2    | 0.1     | 0.0   |

(We assume these metrics are not valuable metrics for OCR-VQA since the ground truth answers are too short in most cases.)

DocVQA:


|           | METEOR | ROUGE-L | CIDEr |
|-----------|--------|---------|-------|
| LLaVA224  | 3.8    | 4.8     | 6.3   |
| LLaVAR224 | 5.6    | 6.9     | 12.7  |
| LLaVA336  | 4.6    | 5.6     | 8.7   |
| LLaVAR336 | 8.6    | 10.0    | 21.5  |

---

### Author Response · Authors · 2023-11-23
**Paper Revision**

Dear reviewers, we have revised the paper to include the new experiment results, the main updates are:
1. Results of extra text-matching evaluation metric: Appendix D and Table 8,9,10,11.
2. More evaluation Scenarios: Appendix E and Table 12.
3. Discussions on the comparison with mPLUG-Owl and the result of finetuning mPLUG-Owl using our data: Appendix F and Table 13.
4. Extended LLaVA architecture with high-resolution encoders: Appendix H, Figure 7, and Table 14.
5. The impact of OCR spelling errors: Appendix I and Table 15.
6. Case study of the recognizable font size: Section 5.3 and Table 6.
7. Ablation study on the LLaVA benchmark: Appendix J and Table 16.

Due to the page limit, most updated content is presented in the Appendix. We’ve added pointers to the Appendix in the main paper.

Please let us know if you have further questions!

---

### Meta-Review · Area_Chair_rVxb · 2023-12-11

**Metareview:**

This paper presents a method to enhance the text-rich image understanding capability of a multimodal LLM, in particular LLaVA. The main contributions of the work are: A new  dataset consists of 400k examples with OCR enhanced text; a new LLaVAR model tuned using the constructed dataset.

The reviewers acknowledge that this work is among the earliest work to dive into visual instruction tuning for text-rich images, and the constructed dataset from this work will surely benefit the community.

However, a primary concern raised by all reviewers is related to novelty. The common understanding is that the distribution of the training dataset significantly impacts performance, and it is expected that incorporating more OCR-ed texts and related tasks would enhance the model's text reading capabilities. Consequently, the overall contribution of the paper is deemed limited.

**Justification For Why Not Higher Score:**

Limited novelty. The contribution of the paper is mostly a new dataset, and the resulting model is expected without any surprise.

**Justification For Why Not Lower Score:**

N/A

---

### Decision · Program_Chairs · 2024-01-16

Reject